

# Vegetation Response to Climatic Variability: Implications for Root Zone Storage and Streamflow Predictions

Nienke Tempel[1], Laurene Bouaziz[2], Riccardo Taormina[1], Ellis van Noppen[1], Jasper Stam[3], Eric Sprokkereef[3] and Markus Hrachowitz[1]

[1]Department of Water Management, Faculty of Civil Engineering and Geosciences, Delft University of Technology, Stevinweg 1, 2628CN Delft, Netherlands
[2]Department Catchment and Urban Hydrology, Deltares, Boussinesqweg 1, 2629 HV Delft, The Netherlands
[3]Ministry of Infrastructure and Water Management, Zuiderwagenplein 2, 8224 AD Lelystad, The Netherlands

*Correspondence to*: Nienke Tempel (nienketessatempel@gmail.com)

**Summary.** This study explores the impact of climatic variability on root zone water storage capacities thus on hydrological predictions. Analysing data from 286 areas in Europe and the US, we found that despite some variations in root zone storage capacity due to changing climatic conditions over multiple decades, these changes are generally minor and have a limited effect on water storage and river flow predictions.

**Abstract.** This paper investigates the influence of multi-decadal climatic variability on the temporal evolution of root zone storage capacities ($S_{r,max}$) and its implications for streamflow predictions at the catchment scale. Through a comprehensive analysis of 286 catchments across Europe and the US, we analyse the deviations in evaporative ratios ($I_E$) from expected values based on catchment aridity ($I_A$) and their subsequent impact on $S_{r,max}$ predictions. Our findings reveal that while catchments do not strictly adhere to their specific parametric Budyko curves over time, the deviations in $I_E$ are generally very minor, with

an average $\Delta I_E = 0.01$ and an interquartile range IQR= -0.01 to 0.03. Consequently, these minor deviations lead to limited changes in predictions of $S_{r,max}$, mostly ranging between -10.5 and +21.5 mm (-5.1% to +9.9%). When these uncertainties in $S_{r,max}$ are incorporated into hydrological models, the impact on streamflow predictions is found to be marginal, with the most significant shifts in monthly evaporation and streamflow not exceeding 4% and 12%, respectively. Our study underscores the utility of parametric Budyko-style equations for first order estimates of future $S_{r,max}$ in hydrological models, even in the face

of climate change and variability. This research contributes to a more nuanced understanding of hydrological responses to changing climatic conditions and offers valuable insights for future climate impact studies in hydrology.

## 1 Introduction

Transpiration from vegetation is, on average, the largest water flux that leaves terrestrial hydrological systems (Jasechko, 2018). In spite of some uncertainty (Coenders-Gerrits et al., 2014), its magnitude is controlled by the interplay between sub-

surface water supply and canopy water demand (Eagleson, 1982; Milly, 1994; Rodriguez-Iturbe et al., 2007; Donohue et al., 2007; Jaramillo et al., 2018). Both, individual plants but also the composition of plant communities within given spatial

 

domains (for brevity hereafter referred to as vegetation), have over the past adapted to local environmental and hydro-climatic conditions to ensure continuous and sufficient access to water, nutrients and light which has allowed their survival (Yuan et al., 2019; Ma et al., 2021). Adaptation strategies include, besides others, the regulation of water use efficiency (e.g. Troch et al., 2009; Flo et al., 2021) or the adaptation of the extent of root systems so that roots penetrate large enough subsurface pore volumes for water supply to satisfy transpiration demand during dry periods (e.g. Gao et al., 2014; Fan et al., 2017). This subsurface pore volume between field capacity and permanent wilting point defines the *maximum* water volume that is within the reach of roots and thus available for plant transpiration, hereafter referred to as root zone storage capacity $S_{r,max}$ (mm). Indeed, $S_{r,max}$ is a core property of terrestrial hydrological systems as it regulates to a large part the partitioning of water fluxes into drainage of liquid water, and thus eventually stream flow Q (mm d$^{-1}$), and vapour released to the atmosphere as transpiration $E_T$ (mm d$^{-1}$) and interception/soil evaporation $E_I$ (mm d$^{-1}$) (Savenije and Hrachowitz, 2017).

At the catchment scale, $S_{r,max}$ has in the past been quantified with three methods. Firstly, by calibration as parameter of hydrological models (e.g. Fenicia et al., 2008; Coxon et al., 2014; Fowler et al., 2020; Bouaziz et al., 2021; Hanus et al., 2021; Wang et al., 2023), secondly as product from estimates of average root depth, soil porosity and water content at field capacity (e.g. Clark et al., 2008; Maxwell et al., 2015) and thirdly by following optimality principles and thus maximizing variables such as net primary production, transpiration rates or others (e.g. Kleidon, 2004; Guswa, 2008; Sivandran and Bras, 2012; Speich et al., 2020). Although all three methods above are correct in principle, insufficient data often limits their use. For example, although there are observations of root-depth of several thousand individual plants worldwide (Guerrero-Ramírez et al., 2021), it is difficult to meaningfully upscale these values to plant communities with different compositions, ages or densities. In addition, these estimates are mostly snapshots in time reflecting past conditions and, similar to the calibration method, do not give any indication about the potential future evolution of $S_{r,max}$.

Alternatively, there is increasing evidence that $S_{r,max}$ can be robustly estimated exclusively based on water balance data, i.e. long-term estimates of precipitation P (mm d$^{-1}$) and actual evaporation $E_A = E_T + E_I$ (e.g. Donohue et al., 2012; Gentine et al., 2012; Gao et al., 2014, 2016; DeBoer-Euser et al., 2016; Wang-Erlandsson et al., 2016; Dralle et al., 2021; Hrachowitz et al, 2021; McCormick et al., 2021; van Oorschot et al., 2021; Stocker et al., 2023). Under the assumption that vegetation allocates resources in an efficient way between above- and sub-surface growth (Guswa, 2008; Schymanski et al., 2008), root systems and thus $S_{r,max}$ will not be larger than necessary to guarantee access to sufficient water during dry periods with certain return periods. The water volume that in the past has been transpired during the driest periods and that can be estimated via the water balance must have been accessible to roots and therefore reflect the magnitude of the water volume that was stored in the sub-surface and accessible to plants during these dry periods, i.e. $S_{r,max}$.

This approach offers the advantage that an evolution of $S_{r,max}$ over time, either through natural adaptation to changing hydro-climatic conditions (e.g. Jaramillo et al., 2018) or through human interventions such as deforestation (e.g. Nijzink et al., 2016a; Hrachowitz, 2021) or irrigation (van Oorschot et al., 2023), is manifest in changes in dry period transpiration $E_T$. This offers an opportunity to not only trace the past evolution of $S_{r,max}$ over time, but together with projections of future hydro-



climatic conditions, including P and $E_P$, also to quantify its potential future trajectories and the associated effects of this
      temporal evolution of $S_{r,max}$ on the hydrological response.

          More specifically, estimating $S_{r,max}$ from the water balance requires knowledge of $E_A$. For past conditions this can be
      robustly estimated from the water balance by assuming negligible storage change, i.e. dS/dt ~ 0, which is satisfied for the vast
      majority of catchments world-wide over time-scales of around 10 years (Han et al., 2020). Climate model projections can
generate, besides estimates of future P and potential evaporation $E_P$ (mm d$^{-1}$), also estimates of future $E_A$. However, the latter
      are subject to major uncertainties (e.g. van Oorschot et al., 2021). As an alternative method, non-parametric formulations of
      the Budyko hypothesis demonstrate that the long-term partitioning of water fluxes, expressed as the evaporative index $I_E$ =
      $E_A/P = 1 - Q/P$ (-), and thus the hydrological response of catchments globally is to the first order controlled by the aridity index
      $I_A = E_P/P$ (Schreiber, 1904; Oldekop, 1911; Budyko, 1948). To reduce the scatter around this non-parametric Budyko-style
curves and to assign catchments a unique position in the $I_A - I_E$ space, parametric re-formulations as for example the Tixeront-
      Fu equation (Tixeront, 1964; Fu, 1981; Zhang et al., 2004) and similar expressions (see Andreassian et al., 2016) were
      developed:

$$\frac{E_A}{P} = 1 + \frac{E_P}{P} - \left[ 1 + \left( \frac{E_P}{P} \right)^\omega \right]^{\frac{1}{\omega}} \qquad (1)$$

where $\omega$ (-) is a catchment specific effective parameter, that aggregates all other influences on $I_E$ next to $I_A$ (Berghuijs and
      Woods, 2016).

          As this relationship has emerged from catchment responses, and thus also vegetation, having adapted to past hydro-climatic
      conditions, expressed by $I_A$, it is plausible to assume that the hydrological partitioning $I_E$ of a catchment will eventually adapt
      to a changing future $I_A$ in a corresponding way by moving along its catchment specific curve defined by $\omega$. This reasoning
then allows to estimate future $E_A$ based on future projections of P and $E_P$ (Roderick and Farquhar, 2011; Wang et al., 2016;
      Liu et al., 2020). As a consequence, the effects of a changing future $E_A$ on the future root zone storage capacity $S_{r,max}$ can be
      quantified. In contrast to the vast majority of climate impact studies, which in the absence of further information, assume time-
      invariant $S_{r,max}$ even under changing future climate (e.g. Prudhomme et al., 2014; Brunner et al., 2019; Hakala et al., 2020;
      Rottler et al., 2020; Hanus et al., 2021), the use of such a time-variant formulation of $S_{r,max}$ as parameter in hydrological models
has the potential to provide more reliable predictions of the future hydrological response of catchments, as for example
      demonstrated in a recent proof-of-concept study by Bouaziz et al. (2022) for the Meuse basin in North-West Europe. They
      found with model simulations that the adaptation of $S_{r,max}$ to future climate conditions, expressed as $I_A$, can cause major shifts
      in seasonal water supply. This involved future increases of $S_{r,max}$, and thus increases vegetation-accessible sub-surface water
      volumes which lead to increases in Summer $E_A$ by up to 15%, which in turn reduced groundwater recharge that resulted in
10% decreases in late-summer and autumn groundwater storage, eventually causing winter flows that can be up to 20% lower
      as compared to model runs that used constant values of $S_{r,max}$ estimated from past hydro-climatic conditions. These findings
      are qualitatively consistent with the results of Speich et al. (2020), who reported significant changes in modelled stream flow





when replacing a static parameter to describe $S_{r,max}$ by a forest dynamics model. More generally, Wagener et al. (2003) and Merz et al. (2011) documented the role of time-variant model parameters including $S_{r,max}$ (in their papers referred to as root constant and FC, respectively) by comparing model calibrations over multiple time windows. In a different approach, the importance of time-variable vegetation dynamics was demonstrated by Duethmann et al. (2020), who used remotely sensed vegetation indices including NDVI to account for temporal variations in evaporation surface resistance in the Penman equation, leading to considerably improved model skill to reproduce observed river flow over multiple decades.

A major assumption underlying the approach of Bouaziz et al. (2022) is that under changing future conditions, catchments will indeed follow their specific Budyko curve as defined by the time-invariant parameter $\omega$, which describes the long-term average past conditions. The resulting $I_E$ is in the following referred to as the expected $I_{E,exp}$. Several recent studies have pointed out that this assumption may not strictly hold and that $\omega$ itself may be subject to fluctuations over time (e.g. Berghuijs and Woods, 2016; Reaver et al., 2022). While of minor relevance for humid environments, $I_E$, and thus $E_A$, becomes proportionally more sensitive to fluctuations in $\omega$ with increasing aridity $I_A$ (Gudmundsson et al., 2016). As a consequence, it has to be expected that estimates of future $E_A$ and the associated $S_{r,max}$, are subject to uncertainties or deviations $\Delta I_{E,exp}$ from $I_{E,exp}$ that are not accounted for by Bouaziz et al. (2022).

The overall objectives of this paper are thus to (1) quantify the uncertainty in $I_E$ following changes in $I_A$ and (2) how this propagates further into uncertainties in time-variant estimates of $S_{r,max}$ in contrasting environments over multiple decades in a large sample approach using long-term water balance data from 286 catchments from the UK, the US and the Meuse basin. In a direct follow up to Bouaziz et al. (2022), who modelled the impact of a changing future climate on the hydrological response in several catchments of the Meuse basin, we will then in a third step, using the same model, (3) quantify the additional effect of uncertainties in $S_{r,max}$ on the hydrological response in the Meuse basin. Specifically, we will test the hypothesis that vegetation adaptation to factors other than $I_A$, manifest in deviations from the expected future $I_{E,exp}$ and thus from the associated future $S_{r,max}$, lead to significant changes in the predicted future hydrological response in the Meuse and need to be accounted for in hydrological climate impact studies.

## 2 Study area & Data

### 2.1 Study area

The hydrological model experiment in this study is done for the Meuse river basin upstream of Borgharen at the border between Belgium and the Netherlands (Figure 1), which spans an area of 21,300 km$^2$ in North-West Europe. To a large part located in the Ardennes, a rolling hill landscape characterized by ridges and incised valleys, the elevation reaches up to around 650m. Approximately 60% of the basin is used for agriculture, while 30% is covered by forests (Bouaziz et al., 2022).

The Meuse basin is characterized by a temperate humid climate with average annual precipitation of around 920 mm yr$^{-1}$, potential evaporation of around 610 mm yr$^{-1}$, and streamflow of around 400 mm yr$^{-1}$. The Meuse is a rain-fed river with a



response time of several hours up to a few days. Transient snow packs can be present for a few days in some parts of the basin but are overall of minor importance (Bouaziz et al., 2021). The streamflow has strong seasonality, with summer low flows and high winter flows, which are on average four times higher than the summer flow (De Wit et al., 2007). Precipitation falls relatively homogenously throughout the year and the seasonality of the streamflow is thus mainly caused by the seasonal differences in solar energy input and thus evaporation.

**2.2 Data**

To quantify the deviations $\Delta I_{E,exp}$ from expected $I_{E,exp}$, we adopted a large sample strategy using long-term water balance data from catchments in contrasting environments.

For the Meuse river basin, daily precipitation, temperature and radiation was obtained for the 1989 – 2018 period from the E-OBS v20.0 dataset (Cornes et al., 2018) and pre-processed as described by Bouaziz et al. (2022). Temperature was 140 downscaled using a digital elevation model and a fixed temperature lapse rate, while potential evaporation was estimated using the Makkink method (Hooghart and Lablans, 1988). Monthly bias correction was applied to address underestimation of precipitation. Daily stream flow data was available for 23 catchments within the Meuse basin from water authorities in Belgium (Service publique de Wallonie), France (Eau France) and the Netherlands (Rijkswaterstaat) for various time periods between 1989 – 2018 (Table 1). Note, that the stream flow data at station Borgharen in the Netherlands is constructed by combining 145 observations from the nearby stations St. Pieter on the Meuse and Kanne on the Albert Canal (De Wit et al., 2007).

Long-term temporal changes in $I_{E,exp}$ and deviations $\Delta I_{E,exp}$ therefrom that can be quantified in the Meuse basin remain limited to the 23 catchments that are gauged and stream flow records of 30 years at most. To increase the sample size in space and time, and to encompass a broader range of climates, we have in addition included data from catchments in the UK and the US, available through the CAMELS GB (Coxon et al., 2020) and CAMELS US (Addor et al., 2017) databases for the time-150 periods indicated in Table 1. To ensure consistency, potential evaporation across all datasets was recalculated using the Makkink equation based on mean daily temperature and shortwave radiation (Hooghart and Lablans, 1988). From the full set of 671 catchments available in each of the two CAMELS databases, we excluded those from the analysis that exhibited long-term $I_E > I_A$, indicating that $E_A$ exceeds $E_P$ and thus the energy limit, which may be an indicator of major data errors or significant unaccounted groundwater export to adjacent catchments (Bouaziz et al., 2018). In addition, catchments were 155 excluded from the analysis if they did not meet the criteria of minimal human impact or received more than 10% of their annual precipitation as snow. The exclusion of catchments with snowfall was necessary due to the temporary water storage capacity of snow, which can lead to inaccurate estimation of root zone storage capacity due to delayed water input (Dralle et al., 2021). This resulted in a total of 286 catchments we have used for the subsequent analysis (23 – Meuse basin; 94 – CAMELS GB; 169 – CAMELS USA; Figure 2) and which cover a wide range of hydro-climatic conditions, with the catchments in the Meuse 160 basin being located at an intermediate position between the UK and US catchments (Figure 3). Finally, the data were segmented into distinct 10-year periods (Table 1), allowing us to quantify decadal changes in $I_A$, $I_E$ and the associated $S_{r,max}$ as well as their decadal deviations from the expected values, i.e. $\Delta I_{E,exp}$ and $\Delta S_{r,max,exp}$.



## 3 Methods

Following the three specific research objectives as formulated in Section 1, the experiment of this study is executed in several subsequent steps, shown in Figure 4: for each of the 286 study catchments (1.1) estimate $I_{E,obs}$, and thus $\omega_{obs}$ and $E_{A,obs}$ from water balance data of multiple past individual decades in the period 1989-2018, (1.2) quantify the distributions of deviations $\Delta I_{E,exp}$ and thus $\Delta E_{A,exp}$ from the expected $I_{E,exp}$ and $E_{A,exp}$ between subsequent decades, (2.1) estimate $S_{r,max,obs}$ from past water balance data and thus from $E_{A,obs}$ of multiple past individual decades, (2.2) quantify the distribution of deviations

$\Delta S_{r,max,exp}$ from the expected $S_{r,max,exp}$ between subsequent decades based on $\Delta E_{A,exp}$. For the 23 Meuse catchments then (3.1) for the 2009-2018 decade sample $S_{r,max,sam}$ from the distribution $\Delta S_{r,max,exp}$ and finally (3.2) quantify the effect of uncertainties in $S_{r,max}$ on the hydrological response by using the sampled values $S_{r,max,sam}$ as parameters in multiple runs and compare the results to model runs that assume $\Delta I_{E,exp}= 0$ and thus $\Delta S_{r,max,exp} = 0$.

### 3.1 Estimate $I_E$, $E_A$ and their deviations from expected values over time

For each of the 286 study catchments we estimated for each individual decade i with a data record (see Table 1) the decadal average evaporation and the associated decadal average evaporative indices from the observed decadal average balance data:

$$E_{A,obs,i} = P_{obs,i} - Q_{obs,i} \tag{2}$$

$$I_{E,obs,i} = E_{A,obs,i}/P_{obs,i} \tag{3}$$

Together with $P_{obs,i}$ and $E_{P,obs,i}$, expressed as aridity index $I_{A,obs,i}$, we then use $E_{A,obs,i}$ to solve the parametric Tixeront-Fu formulation of the Budyko hypothesis (Eq.1) for $\omega_{obs,i}$ for each decade for each individual catchment.

In a next step, assuming that $P_{obs,i+1}$ and $E_{P,obs,i+1}$ of the following decade i+1 are projections of an unknown future, we solve

Eq.(1) for $I_E$ to "predict" the expected evaporative index of that following decade, i.e. $I_{E,exp,i+1} = E_{A,exp,i+1}/P_{obs,i+1}$, based on $I_{A,obs,i+1}$ together with $\omega_{obs,i}$ from the current decade, implying that each catchment will follow its specific curve defined by $\omega_{obs,i}$. The difference of the expected $I_{E,exp,i+1}$ to the actually observed $I_{E,obs,i+1}$ then represents the deviation $\Delta I_{E,exp,i+1}$ and thus $\Delta E_{A,exp,i+1}$ for that decade i+1 for each individual catchment. The deviations for all decades of all catchments are then aggregated to an individual distribution of deviations for each of the three datasets, i.e. $\Delta I_{E,Meuse}$, $\Delta I_{E,UK}$, $\Delta I_{E,US}$ and for one

distribution of all three data sets combined, i.e. $\Delta I_E$. The general procedure is illustrated in Figure 5.





**3.2 Estimate root zone storage capacity $S_{r,max}$ and its deviations from expected values over time**

For each study catchment and each decade i with a data record we estimated the root zone storage capacity $S_{r,max,obs,i}$. This was done on basis of observed decadal water balance data as described elsewhere in detail (e.g. Nijzink et al., 2016a; Bouaziz et al., 2020; Hrachowitz et al., 2021).

Briefly, the decadal averages $E_{A,obs,i}$ (Eq.2) of each study catchment were redistributed to daily values $E_{A,obs,i}(t)$ by rescaling daily observed values of $E_{P,obs,i}(t)$ according to:

$$E_{A,obs,i}(t) = \frac{E_{P,obs,i}(t)}{E_{P,obs,i}} E_{A,obs,i} \qquad (4)$$

where t is any given day within a decade i.

These daily estimates of evaporation $E_{A,obs,i}(t)$ were then used together with daily observed precipitation $P_{obs,i}(t)$ to compute the time series of daily cumulative storage deficits for a specific year j according to:

$$S_{D,j,i}(t) = \begin{cases} \int_{t_0}^{t} \left( P_{obs,i}(t) - E_{A,obs,i}(t) \right) dt, & if \ S_{D,j,i}(t) \leq 0 \\ 0, & if \ S_{D,j,i}(t) > 0 \end{cases} \qquad (5)$$

Where $t_0$ is the last preceding day on which the cumulative storage deficit $S_{D,j,i}(t) = 0$. Note, that the effects of interception evaporation $E_I$ on the estimation of storage deficits are negligible as demonstrated by Bouaziz et al. (2020) and we therefore assumed that $E_A = E_T$.

The maximum annual storage deficit $S_{D,j,i}$ represents the volume of water that needs to be stored within the reach of roots to provide vegetation with continuous access to water in that year j is then obtained as:

$$S_{D,j,i} = max\left( \left| S_{D,j,i}(t) \right| \right) \qquad (6)$$

Previous studies suggested that in a wide spectrum of environments vegetation develops root systems that allow access to sufficient water to bridge dry spells with return periods of around 20 years (Gao et al., 2014; Nijzink et al., 2016a). The annual storage deficits $S_{D,j,i}$ of all years j in a specific decade i and catchment were therefore used to fit Generalized Extreme Value distribution. This then allowed to estimate the storage deficit with a 20-year return period which here was defined as root zone storage capacity for that decade $S_{r,max,obs,i} = S_{D,20yr,i}$.

In a next step, assuming that $P_{obs,i+1}$ of the following decade i+1 is a projection of an unknown future, we follow the same procedure described above by Eqs.(4) – (6) but using $\Delta E_{A,exp,i+1}$ to "predict" the expected root zone storage capacity of the following decade $S_{r,max,exp,i+1}$. The difference between the expected $S_{r,max,exp,i+1}$ and the actually observed $S_{r,max,obs,i+1}$ then represents the deviation $\Delta S_{R,max,exp,i+1}$ for that specific catchment for decade i+1. The deviations for all decades of all catchments





are then aggregated to an individual distribution of deviations for each of the three datasets, i.e. $\Delta S_{r,max,Meuse}$, $\Delta S_{r,max,UK}$, and

$\Delta S_{r,max,US}$.

### 3.3 Effect of $\Delta S_{r,max}$ on stream flow

To isolate and quantify the effect of uncertainties $\Delta S_{r,max}$ in predicted $S_{r,max}$ on predictions of the hydrological response we run several simulation scenarios with a process-based model for the 23 study catchments in the Meuse basin.

**3.3.1 Hydrological model**

The hydrological model used in this study is wflow-FlexTopo (de Boer-Euser, 2017; Verseveld et al., 2022), a fully distributed process-based model designed to represent spatial variability in hydrological processes. The modular model uses flexible model structures for selection of Hydrological Response Units (HRUs), which are delineated based on topography and land use. See Figure 6 for the a schematic representation of wflow-FlexTopo within one HRU.

Briefly, the each HRU consists of several storage components linked by fluxes, similar to comparable process-based models successfully used in previous studies (e.g. Fenicia et al., 2006; Euser et al., 2015; Gao et al., 2016; Fowler et al., 2020). Here, we have defined three HRUs that represent wetlands, hillslopes and plateaus, respectively and which are connected through a common groundwater storage (e.g. Hulsman et al., 2021). The HRUs were delineated using the MERIT hydro dataset at 60 m x 90 m resolution (Yamazaki et al., 2019), with a threshold of 5.9 m for the height above the nearest drainage (HAND; Renno

et al., 2008) and a slope threshold of 0.13, following the methodology proposed by Gharari et al. (2011). The hillslopes are associated with forest and the largest part of plateaus are used for crop cultivation agriculture in the study region, as identified using CORINE land cover data (European Environment Agency, 2018). The areal fraction of each of the three HRUs was derived for each cell at a model resolution of approximately 600 m x 900 m similar to the sub-grid landscape variability implemented by Nijzink et al. (2016b) in the distributed mhM-model. All relevant model equations are given in Table 2. Note

that Horton ponding and Horton runoff processes (Figure 6) have minor importance in the study region and were therefore switch-off for the model implementation in this study.

The model was previously calibrated for at the most downstream gauge at Borgharen (Bouaziz et al., 2022) using a multi-objective calibration strategy based on the Nash-Sutcliffe efficiencies of flows ($E_{NS}$) as well as of the logarithm of flows ($E_{NS,log}$), the Kling-Gupta efficiency of flow ($E_{KG}$) and the monthly runoff coefficients as performance metrics (Bouaziz et al.,

2022). The model was subsequently evaluated for its skill to reproduce stream flow for at all other 22 stream gauges in the Meuse basin on basis of the same performance metrics.





### 3.3.2 Scenarios

The effect of uncertainties $\Delta S_{r,max}$ in predicted $S_{r,max}$ on predictions of the hydrological response in the 23 study catchments

was then quantified by running the calibrated model for the 2009 – 2018 period and replacing $S_{r,max}$ with different "predictions" thereof for that period. Following the procedure to predict $S_{r,max}$ and $\Delta S_{r,max}$ (Section 3.2) from $I_E$ and $\Delta I_E$ (Section 3.1), we have for this experiment used the decadal period 1999 – 2008 ($p_1$) as basis to predict $S_{r,max}$ and $\Delta S_{r,max}$ for the period 2009 – 2018 ($p_2$) in three scenarios.

- *Baseline scenario ($\Delta S_{r,max} = 0$)*: estimate $\omega_{obs,p1}$ of the first decade $p_1$ based on observed data $P_{obs,p1}$, $E_{P,obs,p1}$ and $Q_{obs,p1}$

of that period. Subsequently, we have used these values together with $P_{obs,p2}$ and $E_{P,obs,p2}$ of the second decade $p_2$ to predict the expected $I_{E,exp,p2}$ and finally $S_{r,max,exp,p2}$ for that second period $p_2$ in each catchment. The calibrated $S_{r,max}$ are then replaced in the model by the predicted values $S_{r,max,p2}$. Re-running the model with $S_{r,max,p2}$ for period $p_2$ then provides the baseline output of the hydrological response assuming that the catchments follow their specific Budyko curves as defined by their individual $\omega_{obs,p1}$ and that therefore $\Delta I_{E,p2} = 0$ and $\Delta S_{r,max,p2} = 0$.

- *Scenario A ($\Delta S_{r,max} \neq 0$)*: the catchments do not follow their specific Budyko curves. In this case, we used the combined distributions of historical deviations $\Delta I_E$ from Meuse, UK and US datasets to determine $\Delta S_{r,max,p2}$. The predicted $I_{E,pred,p2}$ for decade $p_2$ was thus estimated by sampling 100 times from the distribution of deviations $\Delta I_E$ and adding the sampled values to the expected value $I_{E,exp,p2}$. This sample of 100 values of $I_{E,pred,p2}$ then allowed to generate a distribution of 100 values $S_{r,max,pred,p2}$ to be used in 100 model re-runs. Thus, each model run represents the effect of an error $\Delta I_E$ in

the estimation of $I_{E,exp,p2}$. The differences in the hydrological responses with respect to the baseline scenario were then quantified (Figure 7a).

- *Scenario B ($\Delta S_{r,max} \neq 0$)*: the same as Scenario A, with the only difference that the not the full distribution of $\Delta I_E$ from all three data sets combined was used. Instead, we have here limited the sampling distribution of deviations to $\Delta I_{E,Meuse}$, and thus to historical deviations in the Meuse basin only to account for potential effects of regionally different

distributions of $\Delta I_E$ (Figure 7b).

## 4 Results

### 4.1 Historical $I_E$ and deviations $\Delta I_E$ from expected values over time

For historical water balance observations of all three datasets, i.e. Meuse, UK and US, the decadal $I_{E,obs}$ exhibits deviations

$\Delta I_E$ from the expected values $I_{E,exp}$ (Figures 8b,d,f, 9), following decadal shifts in $I_A$ with medians of between $\Delta I_A = -0.05$ – 0.11 (Figure 8a,c,e). Overall, the $\Delta I_E$ remains minor, and the distributions largely centre around zero although they do not generally follow Normal distributions as indicated by an analysis of Q-Q plots and Shapiro-Wilk tests for normality (Shapiro and Wilk, 1965). Differences are evident as indicated by non-parametric Wilcoxon rank-sum tests, that suggest significant differences ($p < 0.05$) in the $\Delta I_E$ distributions of the different data sets and different decades. The median $\Delta I_{E,UK}$ for the UK is



rather stable and varies only between 0 and 0.01 with rather narrow spread es shown by the interquartile ranges IQR ~ 0.03. This narrow scatter around zero and the stability over time allow balanced and rather robust predictions of $I_E$ with this data set. On the other hand, the distributions $\Delta I_{E,US}$ for the US catchments are characterized by stronger fluctuations, with medians changing from -0.01 to 0.04 between the decades, and a somewhat wider spread with IQR ~ 0.04.

The noticeable and significant shift towards higher, i.e. more positive $\Delta I_{E,US}$ between these decadal distributions entail

proportionally higher-than-expected evaporation for the later decade. In contrast, while for the first decade the catchments in Meuse basin have a median $\Delta I_{E,Meuse}$ ~ 0  that is broadly consistent with the UK and the US catchments, the distribution experiences a major shift towards lower, i.e. more negative values with a median $\Delta I_{E,Meuse}$ ~ -0.06 in the second decade, suggesting lower-than-expected proportional evaporation. However, this pattern may be an artefact of the limited sample size, consisting of only 9 and 23 catchments, respectively, in the Meuse basin, and should be interpreted with due care to avoid

misinterpretations.

In the deviations $\Delta I_E$ from the expected $I_E$ no apparent geographically pattern can be distinguished from visual analysis (Figure 9). In particular for the UK, adjacent catchments do frequently display opposing signs in $\Delta I_E$, indicating positive and negative deviations from expected $I_E$ within very close distances.

Similarly, aggregating the individual distributions of $\Delta I_E$ of all three data sets and all decades into one full distribution and

stratifying this distribution in to individual distributions according to their aridity index $I_A$ in bins of 0.2 width (Figure 10), also does not exhibit systematic differences between the distributions. The median of all five distributions with $\Delta I_E = 0.00 - 0.01$ close to zero and the spreads are characterized by only minor differences with values of IQR ~ 0.02 for $I_A = 0.2 - 0.4$ and IQR ~ 0.06 for $I_A = 0.8 - 1.0$.

To avoid the need to base the further analysis in the Meuse exclusively on the small sample of $\Delta I_{E,Meuse}$, we initially intended

to construct more robust estimates of $\Delta I_E$ in the Meuse by further stratifying the above according hydro-climatic and landscape indicators. However, this further attempt to find multi-variate relationships that link $\Delta I_E$ with hydro-climatic and landscape indicators did not show clear and consistent results and is not further reported here. As alternative, we therefore decided use two extreme cases of distributions to sample $\Delta I_E$ for the Meuse basins, i.e. Scenario A and Scenario B (Figure 11; Section 3.2.2.), in the subsequent modelling experiment. Based on the results above, the rationale behind using Scenario A is that the

full distribution $\Delta I_E$ describes a large sample of catchments. In the absence of clear pattern of which distribution of deviations $\Delta I_E$ is more suitable for which type of environments, the full distribution, combing all data allows a conservative perspective as it contains a wide range of historically observed $\Delta I_E$ in a wide range of different environments. In addition, note that the full distribution of $\Delta I_E$ from all data with a median $\Delta I_E = 0.01$ and IQR = 0.04 is very similar to the distribution associated with the $I_A$-bin $0.6 - 0.8$ into which most of the Meuse basins fall. The use of the full $\Delta I_E$ distribution in Scenario A is contrasted

by Scenario B and its small sample distribution $\Delta I_{E,Meuse}$. The rationale of Scenario B is to conserve potentially relevant regional information that contained in $\Delta I_{E,Meuse}$ and which may be under-represented in the full distribution due to the differences in the sample sizes.



## 4.2 Historical $S_{r,max}$ and its deviations $\Delta S_{r,max}$ from expected values over time

The general pattern of $S_{r,max}$ estimated from historical water balance data following the procedure described in Section 3.2, is broadly consistent with previous studies (Gao et al., 2014; Wang-Erlandsson et al., 2016; deBoer-Euser et al., 2016; Stocker et al., 2023) and reflects the overall role of hydro-climatic conditions as control in water storage in the root zone of vegetation (Figure 12a). While in catchments in humid regions with low $I_A$ root zone storage capacities as low as $S_{r,max} < 100$mm are predominant, more arid regions with $I_A > 1$ are characterized by significantly higher values of $S_{r,max} > 300$mm. In other words,

vegetation in humid climates has developed smaller root systems due to shorter and less frequent dry spells, which ensure more regular rain water supply which can be directly used for transpiration. In contrast, vegetation in arid climates requires more extensive root systems to access sufficient water throughout the longer and more frequent dry spells.

The changing hydro-climatic conditions between periods $p_1$ and $p_2$, expressed as changes in $I_A$ in Fig. 8, resulted in shifts in $\Delta S_{r,max,obs}$ for $p_2$. Depending on the data set, the median $\Delta S_{r,max,obs}$ was between -24.9 for the Meuse data set and 13.6 mm

for the US data set (Figure 13). The deviations $\Delta I_E$ (Section 4.1) between $p_1$ and $p_2$ then caused corresponding deviations $\Delta S_{r,max,exp}$ from the expected $S_{r,max,exp}$. The results illustrate that the absolute magnitudes and spreads of the deviations from expected root zone storage capacities, i.e. $\Delta S_{r,max,exp}$ remain in general rather limited and closely centred around zero (Figure 1), with a median $\Delta S_{r,max,exp} = 1.39$ mm (IQR = 19.2 mm) in the UK and slightly more pronounced values of $\Delta S_{r,max,exp} = 13.6$ mm (IQR = 43.7 mm) in the US. The relative deviations show a similar picture with medians of 0.8% (IQR = 11.9%) in the

UK and 4.8% (IQR = 16.5%) in the US. Reflecting the higher $\Delta I_{E,Meuse}$, $\Delta S_{r,max,exp}$ with a median of -24.9 mm is characterized by more marked negative deviations in the Meuse basin, suggesting that the expected root zone storage capacity is overestimated and therefore smaller than expected.

As shown in Figure 12, both the absolute and relative magnitudes of $\Delta S_{r,max,exp}$ do not show a clear relationship with $I_A$. Throughout all types of environments, from humid to arid, most deviations $\Delta S_{r,max,exp}$ remain closely confined to the range

-25 – 25 mm or -5% - 5% in relative terms. The only exception are that the highest positive and negative $\Delta S_{r,max,exp}$ occur in the aridity 0.75 – 1.0 aridity bin, with values reaching extreme values of -124 mm and + 147 mm, which may however also be a mere artefact of the considerably larger sample of catchments in this aridity zones than for more humid or more arid regions.

## 4.3 Effect of $\Delta S_{r,max}$ on stream flow predictions

### 4.3.1 Overall model performance

The model calibrated to observed stream flow at station Borgharen at the outlet of the Meuse basin captures the main features of the hydrological response at that location. Slightly underestimating low flows and overestimating a few peaks, such as in January 2011, the model performance at Borgharen was obtained as $E_{NS} = 0.85$, $E_{NS,log} = 0.72$ and $E_{KG} = 0.88$ (Figure 14a). This is mirrored by the model's ability to reproduce stream flow in the remaining 22 sub-catchments (Figure 14b,c),

which largely exhibit only moderately lower performances with median $E_{NS} = 0.72$, $E_{NS,log} = 0.75$ and $E_{KG} = 0.80$ (Figure 14d).



However, for two of the catchments (Modave and Jemelle) the model could not well reproduce the hydrological response. The underlying geology of these catchments is complex and they are likely experiencing major groundwater losses which are not accounted for in this model (Bouaziz et al., 2018).

### 4.3.2 Changes in the hydrological response due to $\Delta S_{r,max}$ – Scenario A

Sampling from the full distribution $\Delta I_E$ as described in Section 3.3.2, and re-running the model for period $p_2$ with the associated values $\Delta S_{r,max}$ the resulting modelled evaporation and stream flow were compared to that of the baseline scenario ($\Delta I_E = 0$ and $\Delta S_{r,max} = 0$). Overall, it was found that the modelled annual average evaporation and stream flow was affected only to a minor degree by $\Delta S_{r,max}$, with median values of between $\Delta E_A \sim 0.00 - 5.08$ mm yr$^{-1}$ ($< 1\%$) and $\Delta Q \sim -6.55 - 0.63$ mm yr$^{-1}$ (-1.32% - 0.31%) for all catchments.

However, minor but distinguishable shifts in seasonal re-distribution of water fluxes could be observed. Early summer to early autumn $E_A$ increases on average by up to $-1.97 - 3.69$ mm month$^{-1}$ with an IQR = $-0.64 - 1.83$ mm month$^{-1}$, depending on the catchment. In relative terms this is equivalent to increases of up to -2.47% - 4.71% with IQR = $-0.77 - 2.27\%$ (Figure 15a). More specifically, at station Borgharen, $\Delta S_{r,max}$ causes the highest annual change in June with a median $\Delta E_A = 0.39$ mm month$^{-1}$ (0.48%) with an IQR = $0.04 - 0.78$ mm month$^{-1}$ (0.05% – 0.95%; Figure 15b). Similar changes can be observed in other catchments (Figure 14c, d and Supplementary Material Figures S5 – S27). Higher summer $E_A$ due to $\Delta S_{r,max}$ is contrasted by reduced winter stream flow, which generally reaches on average $\Delta Q$ by up to $-4.40 - 2.37$ mm month$^{-1}$ (-0.51% – 7.33%) with IQR = $-2.27 - 0.74$ mm month$^{-1}$ (-5.84 – 2.47%; Figure 15e). At Borgharen, the most pronounced changes occur in December with median $\Delta Q = -0.09$ mm month$^{-1}$ (-1.04%) and IQR = $-0.90 - 1.34$ mm month$^{-1}$ (-2.24% – -0.10%; Figure 15f).

Analysing the effect of $\Delta S_{r,max}$ on modelled annual maximum flow $\Delta Q_{max}$, it can be seen that across all 23 study catchments in the Meuse basin, changes also remain rather limited, with a median $\Delta Q_{max} = -0.06$ mm d$^{-1}$ ($< 1\%$) and IQR = $-0.20 - 0.07$ mm d$^{-1}$ (-2.45% – 0.80%; Figure 16a). The most pronounced $\Delta Q_{max} = -0.29$ mm d$^{-1}$ (-1.28%) with IQR = $-0.39 - 0.07$ mm d$^{-1}$ (-4.37% – 0.80%) was observed in the Le Mouzon Circourt-sur-Mouzon catchment, while at Borgharen $\Delta Q_{max} = -0.07$ mm d$^{-1}$ (-0.91%) was found (Figure 16b). The opposite can be observed for annual minimum flows, which experienced a slight overall increase of $\Delta Q_{min} = 0.0002$ mm d$^{-1}$ ($< 1\%$) with IQR = $-0.0002 - 0.001$ mm d$^{-1}$ caused by $\Delta S_{r,max}$ (Figure 16a). The modelled $\Delta Q_{min}$ at Borgharen reached 0.0005 mm d$^{-1}$ (-0.30%) with IQR = $0.0002 - 0.0009$ mm d$^{-1}$ (Figure 15b).

### 4.3.3 Changes in the hydrological response due to $\Delta S_{r,max}$ – Scenario B

Alternatively, sampling from the sparse distribution $\Delta I_{E,Meuse}$ as described in Section 3.3.2 to estimate $\Delta S_{r,max}$ for model re-runs, provided a perspective on how more extreme, regionally confined distributions of $\Delta I_E$ may affect the hydrological response. Similar to Scenario A, the modelled average annual changes of $\Delta E_A \sim -38.2 - 0.03$ mm yr$^{-1}$ (-4.08% – 0.02%) and





$\Delta Q \sim$ -2.59 – 43.98 mm yr$^{-1}$ (-1.06% – 12.48%) remained modest across all study catchments in the Meuse basin, albeit slightly more pronounced than for Scenario A.

In contrast, major differences were detected in the modelled seasonal water fluxes. On average, catchments experienced a

reduction of summer evaporation, in particular in the months June and July, with $\Delta E_A$ = -10.73 – 0.02 mm month$^{-1}$ (-12.79% - 0.20%) and IQR = -6.00 – 1.04 mm month$^{-1}$ (-8.16% – -1.28%) as shown in Figure 17a. Zooming in to the stations Borgharen, Ortho and Chooz, corresponding pattern of $\Delta E_A$ can be found (Figures 17b-d) with the most pronounced $\Delta E_A$ = -0.56 mm month$^{-1}$ (-1.28%) and IQR = -11.39 – 0.08 mm month$^{-1}$ (-13.8 – 0.53%) at station La Meuse Goncourt (Supplementary Material Figure S4). Seasonal stream flow experienced partly considerable increases. The modelled increases $\Delta Q$ were, on average,

most pronounced in late autumn and early winter across all catchments, with median values of up to 3.74 mm month$^{-1}$ (8.53%) and IQR = 1.21 – 7.63 mm month$^{-1}$ (2.83 – 7.36%) in December (Figure 17e). At Borgharen $\Delta Q$ = -1.98 – 3.86 mm month$^{-1}$ (5.01 – 13.05%) with IQR = 0.60 – 2.12 mm month$^{-1}$ (6.93 – 11.48%) was found (Figure 17f).

The modelled annual maximum flows increased through for Scenario B with a median $\Delta Q_{max}$ = 0.33 mm d$^{-1}$ (4.8%) and IQR = 0.10 – 0.71 mm d$^{-1}$ (1.43% – 8.76%) across all study catchments in the Meuse basin (Figure 16a). The most pronounced

$\Delta Q_{max}$ = 0.96 mm d$^{-1}$ (6.08%) with IQR = 0.18 – 1.69 mm d$^{-1}$ (1.11 – 17.44%) was observed in the La Meuse Goncourt catchment, while at Borgharen $\Delta Q_{max}$ = 0.17 mm d$^{-1}$ (5.41%) was found (Figure 16b). In spite of these partly marked increases in $Q_{max}$, the effect of $\Delta S_{r,max}$ on annual minimum flows in Scenario B remained low and comparable to those from Scenario A. For all catchments $\Delta Q_{min}$ was found to be close to zero, with a median $\Delta Q_{min}$ = -0.0001 mm d$^{-1}$ (< -1%) and IQR = -0.0003– 0.0005 mm d$^{-1}$ (Figure 16a). More specifically the modelled $\Delta Q_{min}$ at Borgharen reached 0.001 mm d$^{-1}$ (0.8%) with IQR =

0.0008 – 0.002 mm d$^{-1}$ (Figure 16b).

## 5 Discussion

Parametric formulations of the Budyko hypothesis, such as the Tixeront-Fu equation (Eq.1; Tixeront, 1964; Fu, 1981) have in the past been used to predict $I_E$ and thus future water partitioning based on changes in $I_A$ under the assumption that

catchments follow their specific curves in the $I_A$-$I_E$ space, as defined by parameter $\omega$ that is obtained from long-term historical water balance data (Roderick and Farquhar, 2011; Wang et al., 2016; Liu et al., 2020). Recently, several studies correctly observed that catchments do not necessarily follow their specific curves under changing environmental conditions raising the concern that parametric Budyko-style equations may therefore have little predictive power (Berghuijs and Woods, 2016; Reaver et al., 2022; Jaramillo et al., 2022). Our results indeed provide further evidence that such deviations $\Delta I_E$ from expected

$I_E$ are a widespread phenomenon. However, our results also illustrate that, although catchments do not strictly follow their specific curves at decadal time scales, the magnitude of deviations remains, overall, rather minor with a median of $\Delta I_E$ = 0.01 and an IQR = -0.01 – 0.03 across all catchments in this study (Figure 10). In spite of some differences in detail, the general distributions of $\Delta I_E$ from different datasets, regions and hydro-climatic conditions are broadly similar and no systematic differences linked to catchment properties or hydro-climatic conditions could be identified (Figures 9, 10).



The root zone storage capacity $S_{r,max}$ as a core property of terrestrial hydrological systems and parameter in hydrological models, can together with its evolution over time be robustly estimated at the catchment-scale based on water balance data (Gao et al., 2014; DeBoer-Euser et al., 2016; Wang-Erlandsson et al., 2016; Dralle et al., 2021; Hrachowitz et al, 2021; McCormick et al., 2021; van Oorschot et al., 2021, 2023; Stocker et al., 2023). This offers an opportunity to account for vegetation adaptation to changing hydro-climatic conditions with a time-variable parameter $S_{r,max}$ for predictions with

hydrological models. Bouaziz et al. (2022) were the first to demonstrate the potential of doing that in a recent proof-of-concept study. However, they estimated future $S_{r,max}$ under the assumption that their catchments will *strictly follow* their specific curves in the $I_A - I_E$ space as determined by parameter ω which was obtained from historical water balance data. In other words, they did not account for deviations $\Delta S_{r,max}$ that result from deviations $\Delta I_E$. Addressing this knowledge gap, we here found that for the vast majority of 286 catchments analysed in this study, characterized by a median historical $S_{r,max} = 239.2$ mm, the limited

deviations $\Delta I_E$ also resulted in $\Delta S_{r,max}$ that remained narrowly confined between ~ -10.5 – 21.5 mm or -5.1 – 9.9 % (Figure 13), although some few regional outliers can reach higher values.

Using samples of $\Delta I_E$ from two distinct distributions in Scenarios A and B we estimated $\Delta S_{r,max}$ for use as parameter in model simulations. Overall it was found with the more balanced Scenario A that $\Delta S_{r,max}$ caused shifts in seasonal $E_A$ and Q, however, characterized by marginal magnitudes with the most pronounced changes $\Delta E_A < 1$ %, on average, occurring in June

and for $\Delta Q \sim -1\%$ in December, with similar pattern for the annual maximum and minimum flows, $\Delta Q_{max}$ and $\Delta Q_{min}$, respectively.  the more equilibrated Scenario A. For Scenario B, an example of a rather extreme, regionally confined $\Delta I_E$, the deviations showed somewhat higher magnitudes with $\Delta E_A \sim -4$ % in July and $\Delta Q \sim 12\%$ in November and comparable pattern for $\Delta Q_{max}$ and $\Delta Q_{min}$.

Notwithstanding the above, it is important to bear in mind that, as in any catchment-scale hydrological experiment, the

available data may be subject to various types of uncertainties, which can be further exacerbated by decisions in the modelling process (e.g. Beven, 2016; Nearing et al., 2016; Hrachowitz and Clark, 2017; McMillan et al., 2018) so that results have to be interpreted with due care. This is in particular true for the use of long time-series of data records generated by different data providers, potentially also using changing observation methods over time and for which, in many cases, homogenization to make them comparable is not a trivial task. In this study, the use of E-OBS precipitation data together with stream flow data

from various data providers in Belgium, France and the Netherlands for the Meuse basin, as well as the CAMELS GB and US datasets initial analysis illustrated the presence of systematic differences in the water balances between the three individual groups of data sets. These differences could in a preliminary analysis here be largely attributed to different methods to estimate $E_P$. While for the data record of the Meuse basin, the lack of more detailed consistent long-term data dictated the use of the Makkink equation based on temperature and incoming short-wave radiation, $E_P$ was estimated using the Penman-Monteith

method in the CAMELS GB catchments and the Priestley-Taylor method in the CAMELS US catchments, respectively. In an attempt to homogenize across the data sets we therefore re-estimated $E_P$ in the UK and US catchments with the Makkink equation. Its simplicity and the exclusion of factors such as vapour pressure deficit or wind speed may have the potential to





cause a certain level uncertainty, although it has previously been shown to produce plausible estimates of $E_P$ for use in hydrological models (Oudin et al., 2005).

Another unresolved issue that emerged from our analysis is the considerable reduction of evaporation in the Meuse basin between the two study periods $p_1$ (1999 – 2008) and $p_2$ (2009 – 2018), as illustrated by the distribution of $\Delta I_E$ that is characterized by remarkably more negative bias (Figure 9) than in any other study catchment. The origin of this pattern is unclear, but similar anomalies in the hydrological response have previously been reported for the mid-20th century by others (Fenicia et al., 2009). They put forward the hypothesis that major decadal fluctuations of $I_E$ in the Meuse basin may have been

the result of active, large scale forest management. More specifically, forest rotation and a shift from deciduous to coniferous forest together with an increase in average forest age towards the end of the 20th century was hypothesized to have caused $I_E$ fluctuations observed in the Meuse basin. While the relationship between stand age and evaporation is still under investigation (Teuling and Hoek van Dijke, 2020), there is evidence that young forests tend to evaporate more than mature forests (e.g. Vertessy, 2001; Brown et al., 2005). Together with Dirkse and Daamen (2004) who noted that in the Netherlands, changes in

forest management practices from clear-cutting and increased thinning resulted in a 10-year increase in the average age of trees between 1980 and 2001, from 43.3 to 53.3 years, this may indeed explain at least some of the $\Delta I_E$. observed in the Meuse basin, although it remains unclear why similar pattern were not observed elsewhere.

Together, the results of this study suggest that although most catchments do not strictly follow their specific curves in the $I_A – I_E$ space over time, the general magnitudes of deviations $\Delta I_E$ are in general low enough to cause only very minor deviations

$\Delta S_{r,max}$ in predictions of root zone storage capacities. As a consequence, even under the assumption of rather exceptional $\Delta I_E$ and thus $\Delta S_{r,max}$ in Scenario B, the effects on the hydrological response remain limited. This further suggests that vegetation adaptation to factors other than $I_A$ and which are manifest in the deviations $\Delta I_E$ and eventually in $\Delta S_{r,max}$, does overall not lead to major changes in the predicted future hydrological response in the Meuse. In spite of not strictly following their specific curves, catchment estimates of future $I_E$, based on changes in future $I_A$ may therefore still be considered useful as first order

estimates to quantify the future evolution of parameter $S_{r,max}$ in hydrological models for climate impact studies over decadal time scales.

## 6 Conclusions

In this study we have quantified the cascading effects of uncertainties in decadal predictions of evaporative ratios $I_E$ as a

function of changes in catchment aridity $I_A$ on predictions of root zone storage capacities $S_{r,max}$ and eventually on predictions of stream flow. In a large sample study, involving long-term data from 286 catchments in Europe and the US, it was found (1) that catchments do not strictly follow their specific curves defined by parameter $\omega$ in the $I_A – I_E$ Budyko space over multiple decades, but these deviations are characterized by limited magnitudes with, on average, $\Delta I_E =$ 0.01 (0.89%) and IQR = -0.01-0.03 (-2.28% – 4.84%), (2) that the deviations $\Delta I_E$ have a minor impact on predictions of $S_{r,max}$ with the resulting deviations

$\Delta S_{r,max}$ ranging mostly between -10.5 – +21.5 mm or -5.1 – +9.9 % finally (3) that these uncertainties $\Delta S_{r,max}$ have only limited



effect on the hydrological response: in spite of causing shifts in seasonal water supply, the magnitudes of these shifts in monthly $E_A$ and Q largely remain very minor (< 1%) and do not, even in the exceptional Scenario B, exceed 4% ($E_A$) to 12% (Q). Overall, this suggests that uncertainties in predictions of $I_E$ based on parametric Budyko-style equations and the associated uncertainties in predictions of model parameter $S_{r,max}$ do not cause major uncertainties in stream flow predictions and can thus
be considered useful first order estimates in the absence of more detailed information.

*Competing interests:* At least one of the (co-)authors is a member of the editorial board of Hydrology and Earth System Sciences.

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





List of Figures



(a)                                              (b)

Figure 1: (a) The location of the Meuse basin in North-West Europe, where the red color indicates the aridity index ($I_A$) of the catchment, (b) The elevation range, river trajectory and gauges in the Meuse river basin. Gauges are indicated with orange dots. The catchments of Borgharen, Ortho, and Chooz are specifically highlighted (with green), as they are separately emphasized in some of the results and analyses.



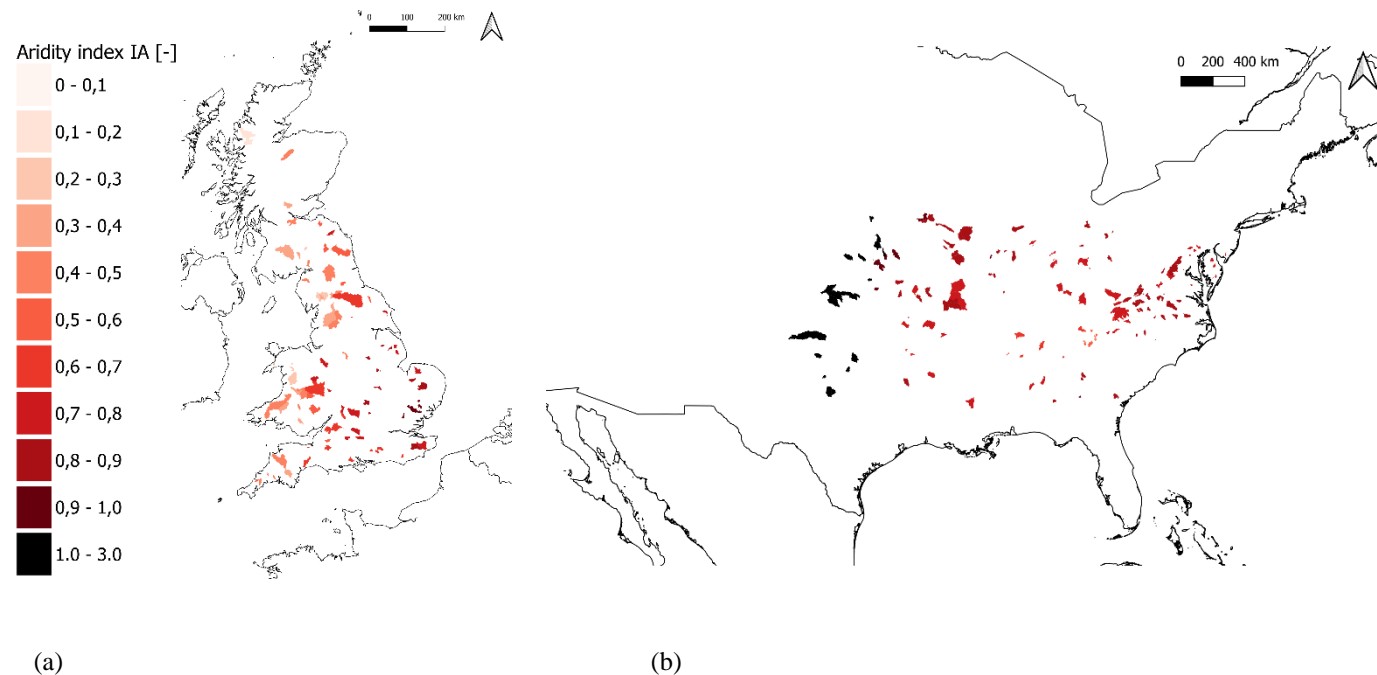

(a)                                                                          (b)

Figure 2: The locations of the catchments that are provided by the large sample datasets (a) CAMELS GB, (b) CAMELS USA. The red color indicates the aridity index ($I_A$) of the catchment.



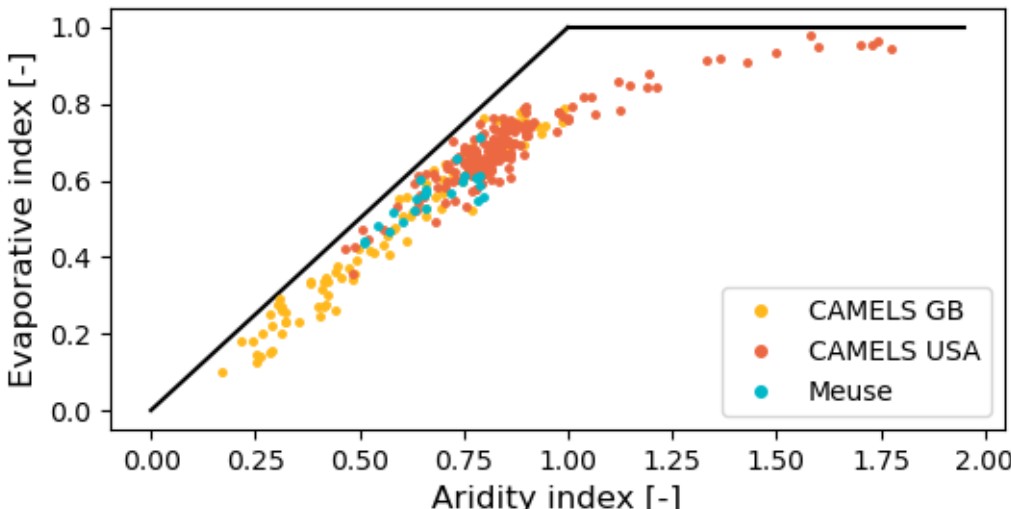

Figure 3: Illustration of the long-term average values for the entire period of data analysis.



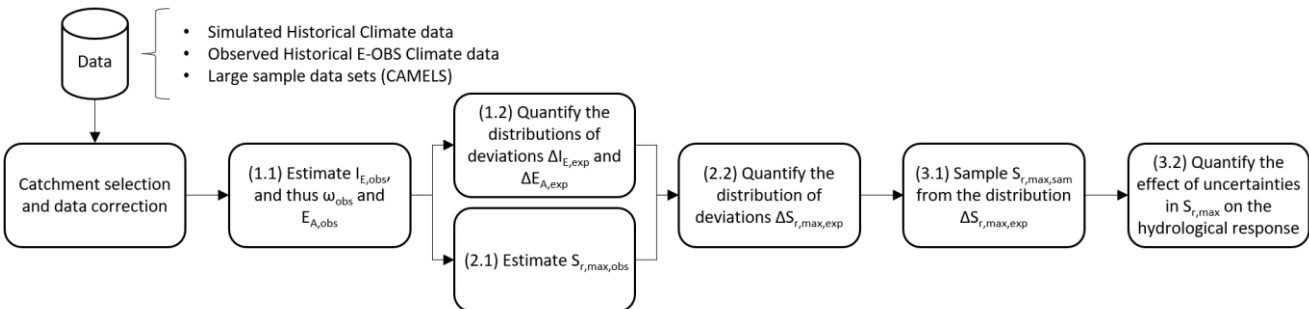

Figure 4: Overview of the methodological procedure.





Figure 5: The step-by-step process for calculating the error in $I_E$ ($\Delta I_E$) using data with three decades as an example.



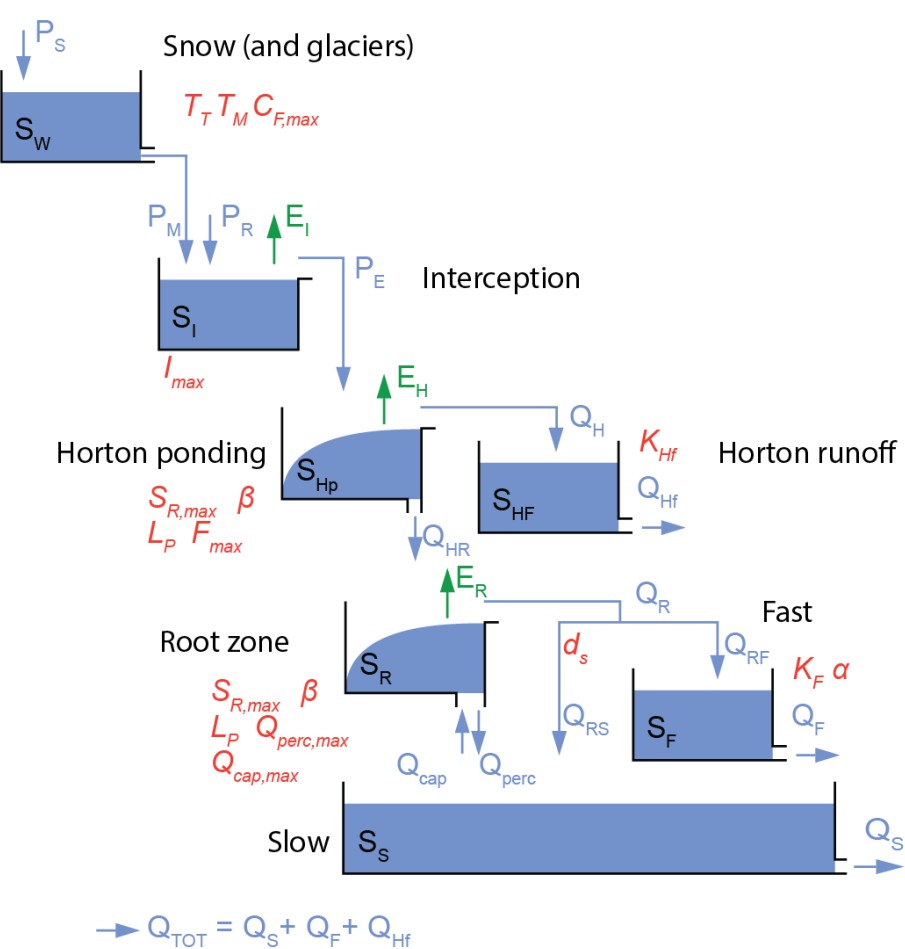

Figure 6: Schematic representation of the wflow FlexTopo model for a single class model including all storages and fluxes (Verseveld et al., 2022).





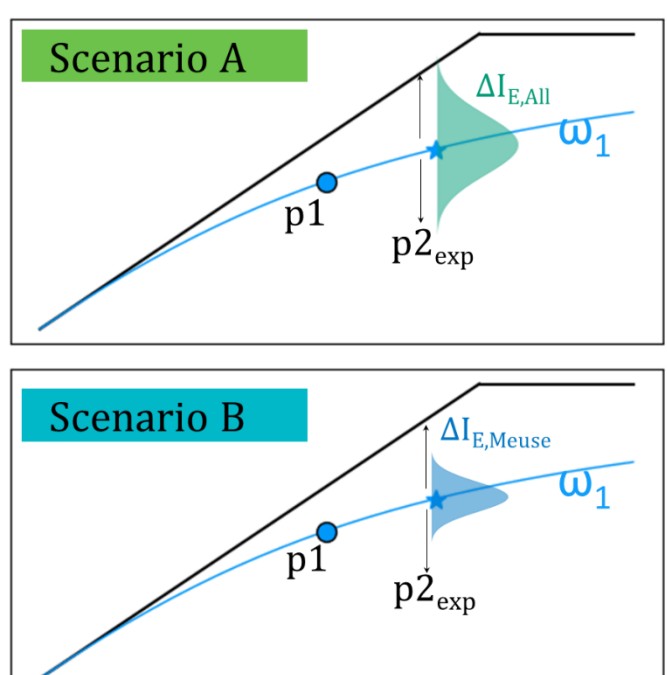

Figure 7: Overview of the scenario structure. p1 is the time period of 1999-2008 and p2 of 2009-2018.





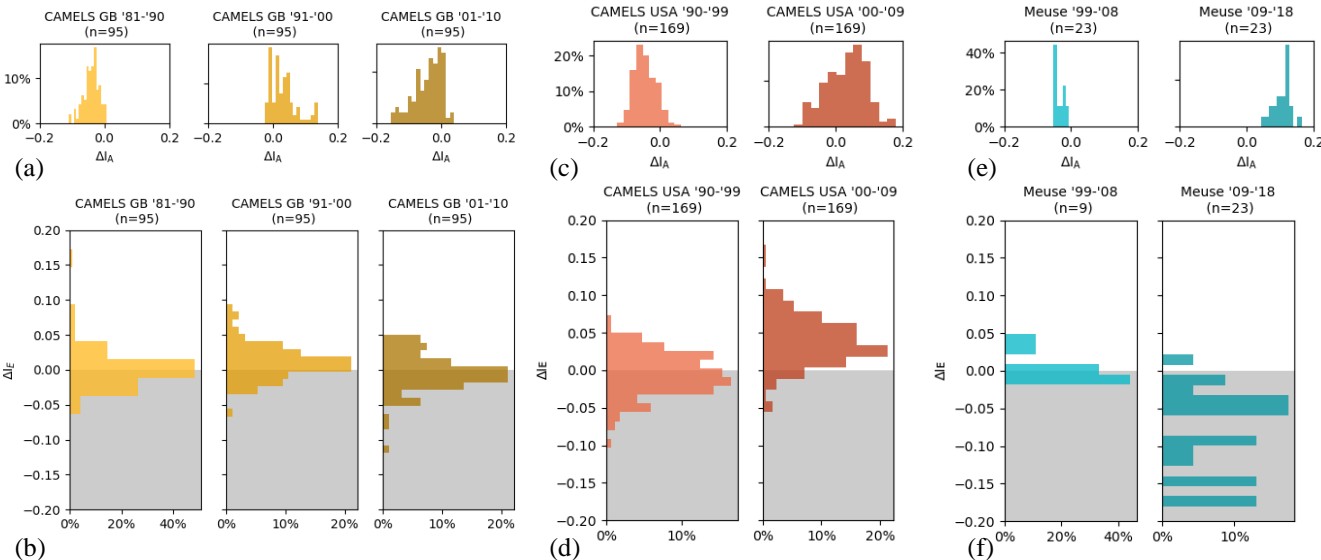

Figure 8: Distributions of $\Delta I_A$ per decade and per dataset CAMELS GB (a), CAMELS USA (c) and Meuse (e), and the deviations in estimating $I_E$ ($\Delta I_E$) for the different datasets CAMELS GB (b), CAMELS USA (d) and Meuse (f).



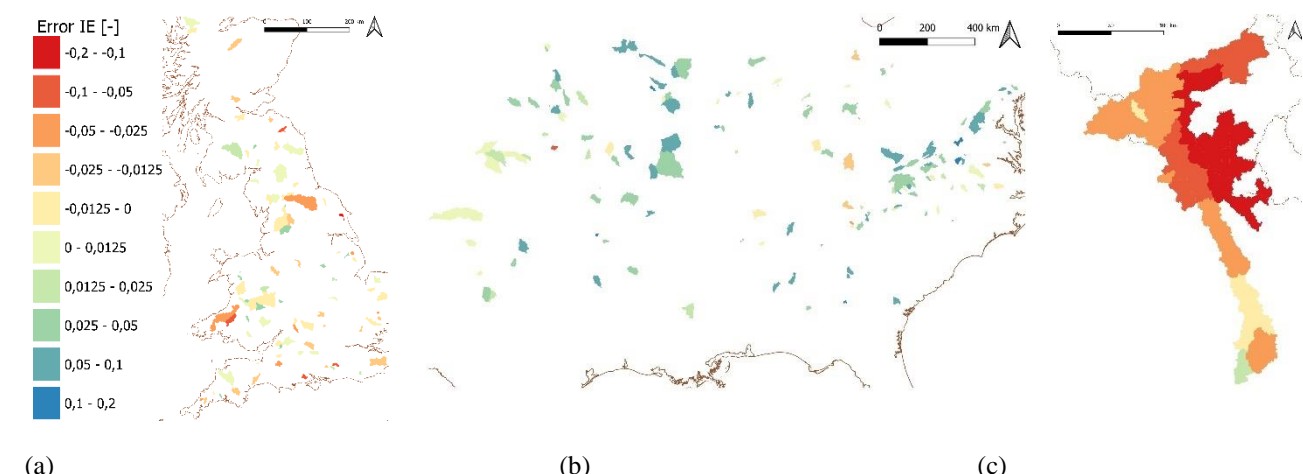

(a)                              (b)                              (c)

Figure 9: Errors in estimating $I_E$ for different time periods and regions: (a) GB for the period 2001-2010, (b) USA for the period 2001-2010, and (c) Meuse for the period 2009-2018. The colours indicate the error in $I_E$ ($\Delta I_E$), ranging from -0.2 (red) to +0.2 (blue). Note that the colour scale is unevenly distributed to emphasize differences around zero.







Figure 10: (a) $I_E$ deviations ($\Delta I_E$) plotted per aridity index in the Budyko framework, for all datasets combined,

(b) Distributions of $I_E$ deviations ($\Delta I_E$) corresponding to the aridity index groups from (a), (c) Distributions of

$I_E$ deviations ($\Delta I_E$) for all data combined and Meuse data.



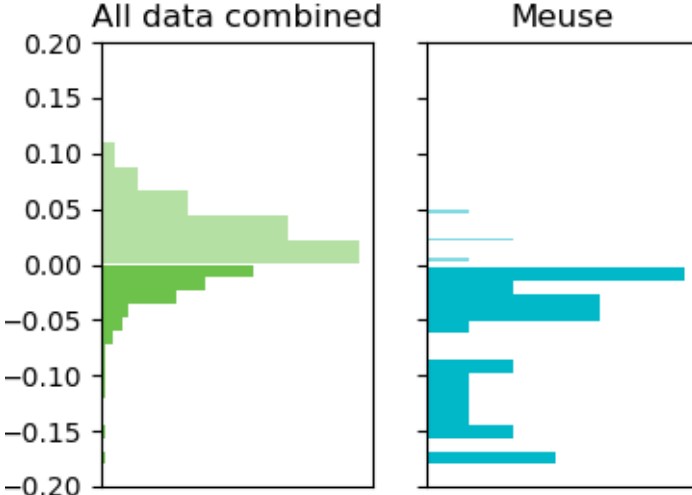

Figure 11: The distributions in deviations from $I_E$ ($\Delta I_E$) used for the samples.



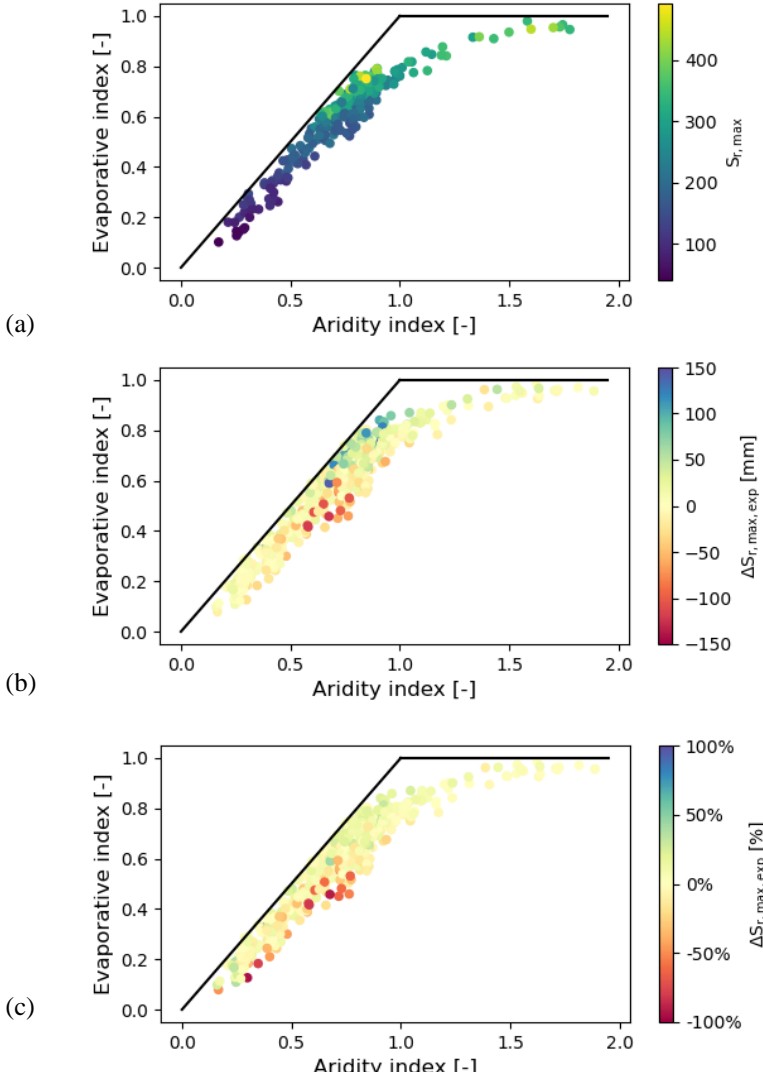

Figure 12: (a) Root zone storage capacities (Sr,max) in the Budyko framework for Meuse, CAMELS GB, and CAMELS USA catchments. (b) and (c) show the error in estimating the root zone storage capacity (ΔSr,max,exp) plotted in the Budyko framework by colour scale, in absolute values [mm] and in percentages [%] respectively.




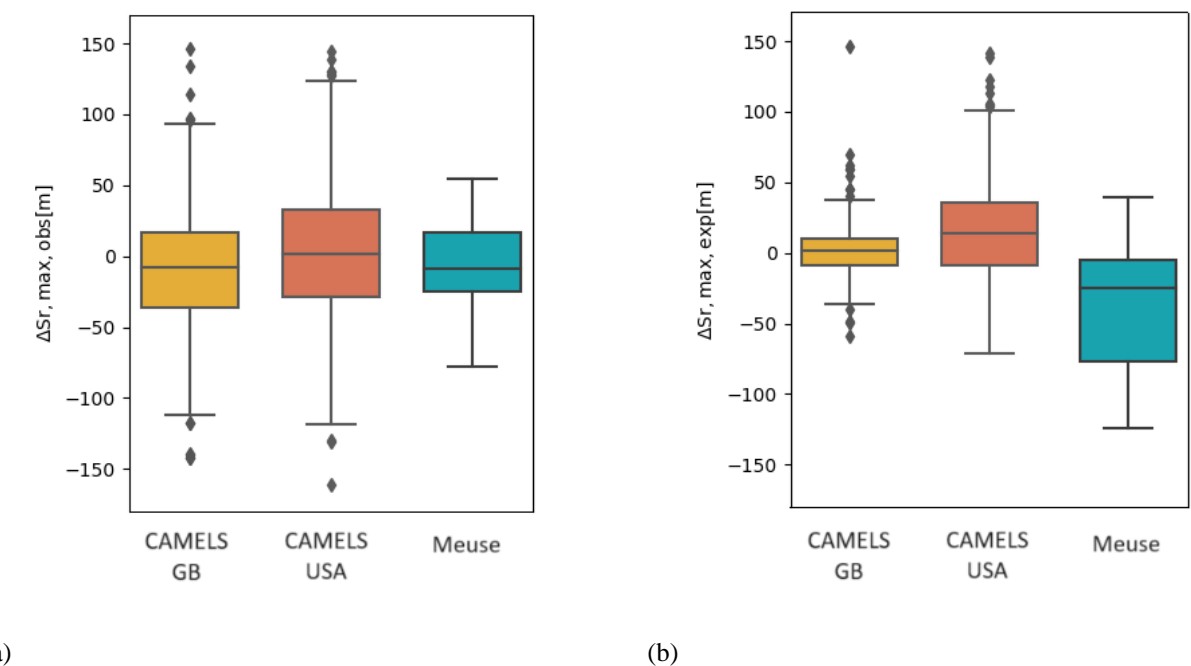

(a)                                              (b)

Figure 13: The distributions of shifts in (a) $\Delta S_{r,max,obs}$, and (b) $\Delta S_{r,max}$ for the different datasets.





Figure 14: Performance measurement through (a) hydrograph of modeled and observed streamflow in catchment Borgharen, (b) Ortho, (c) Chooz and (d) performance indicators calculated for the time period of each scenario: Nash-Sutcliffe efficiencies of streamflow (NSE), the logarithm of streamflow (NSElog), and Kling-Gupta efficiency of streamflow (KGE). The bandwidth of each violin represents the distribution of performance across different catchments and the catchments from (a), (b) and (c) are indicated.





Figure 15: Change in evaporation and streamflow for scenario A. The change is calculated for every run as the difference between the evaporation (a-d) or streamflow (e-h) with the reference run ($\Delta I_E = 0$). The output for all years, and runs has been put together. The lightly shaded area represents the 90th and 10th percentiles, while the slightly darker shaded area represents the 25th to 75th percentiles. The black line represents the median. Images (a) and (e) display all catchments, (b) and (f) Borgharen, (c) and (g) Ortho and (d) and (h) Chooz.



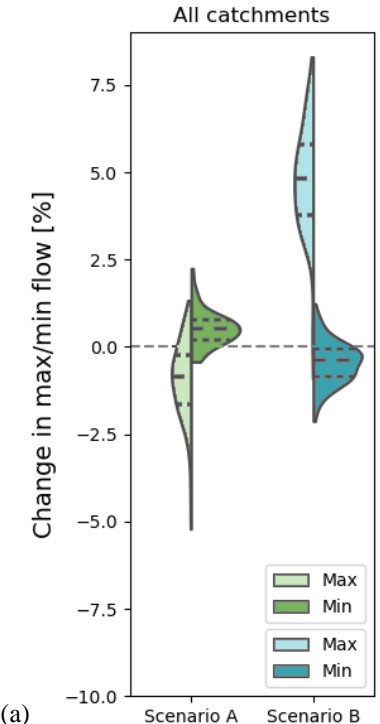

(a)

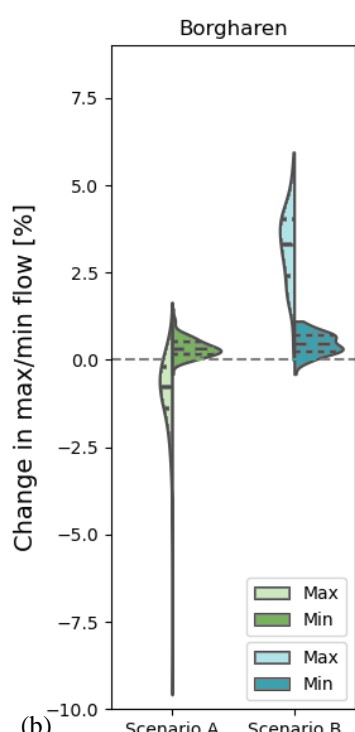

(b)

Figure 16: Change in maximum flow (Qmax, left part of the violin) and 7-day minimum flow (Qmin, right part of the violin), in percentage of the reference run. The quartiles are indicated with dashed lines, for (a) all catchments and (b) Borgharen





Figure 17: Change in evaporation and streamflow for scenario B. The change is calculated for every run as the difference between the evaporation (a-d) or streamflow (e-h) with the reference run ($\Delta I_E = 0$). The output for all years, and runs has been put together. The lightly shaded area represents the 90th and 10th percentiles, while the slightly darker shaded area represents the 25th to 75th percentiles. The black line represents the median.



Table 1: Segmentation of data by 10-year periods, with exception of the CAMELS USA which has two periods of 9

years. Note the extra time period for the France Meuse data, in comparison with the Belgium/Netherlands data.

| Dataset | Data periods (1ST Jan. first year – 31ST dec. last year) | | | | |
|---|---|---|---|---|---|
| Camels GB | 1971 – 1980 | 1981 – 1990 | 1991 – 2000 | 2001 – 2010 | |
| Camels USA | | 1981 – 1989 | 1990 – 1999 | 2000 – 2009 | |
| Meuse Belgium and The Netherlands | | | | 1999 – 2008 | 2009 – 2018 |
| Meuse France | | | 1989 – 1998 | 1999 – 2008 | 2009 – 2018 |

760





Table 2: Water balance and flux equations used in the hydrological model, with variables: $P_S$ is Snowfall [mm t$^{-1}$], $Q_M$ is Snowmelt [mm t$^{-1}$], $Q_R$ is Refreezing snow [mm t$^{-1}$], $T_{thresh}$ is Melting temperature threshold [ºC], $T_{range}$ is the range over which precipitation is partly falling as snow and partly as rainfall [ºC], T is Air temperature [ºC], $Q_{R,pot}$ is Potential Snowmelt [mm t$^{-1}$], $s_{DDF}$ is Degree-day factor [mm t$^{-1}$ ºC], $s_{RF}$ is a coefficient of refreezing [-], $P_R$ is Rainfall [mm t$^{-1}$], $I_{max}$ is Maximum interception storage for each class [mm], $E_I$ is Interception evaporation [mm t$^{-1}$], $P_E$ is Effective precipitation [mm t$^{-1}$], $L_P$ is Threshold parameter for water stress [-], $F_{max}$ is Maximum infiltration capacity [mm t$^{-1}$], $F_{dec}$ is Decay coefficient [-], $S_{R,max}$ is the root zone storage capacity [mm], $Q_{R,direct}$ is Direct runoff [mm t$^{-1}$], $Q_{R,in,net}$ is Net infiltration in the root zone storage [mm t-1], $E_R$ is Evaporation from the root zone storage [mm t$^{-1}$], $Q_R$ is Runoff [mm t$^{-1}$], $Q_{perc}$ is Percolation to the slow groundwater [mm t$^{-1}$], $Q_{cap}$ is Capillary rise from the slow groundwater [mm t$^{-1}$], $Q_{perc,max}$ is Maximum percolation parameter [mm t$^{-1}$], $Q_{cap,max}$ is Maximum capillary rise flux parameter [mm t$^{-1}$], $Q_{RF}$ is Inflow in the fast storage [mm t$^{-1}$], $Q_F$ is Fast runoff [mm t$^{-1}$], $K_F$ is Recession constant [t$^{-1}$], $Q_{RS}$ is Preferential recharche from the outflow of the root zone storage [mm t$^{-1}$], $Q_S$ is Linear outflow from the slow groundwater storage [mm t$^{-1}$], $K_S$ is Recession timescale coefficient, $Q_{TOT}$ is Total streamflow [mm t$^{-1}$], $F_{hrufrac}$ is Fraction of each class in a cell [-].

| Storage component | Water balance | Constitutive equations |
|---|---|---|
| Snow storage | $\dfrac{dS_w}{dt} = P_S - Q_M + Q_R$ | $P_S = P * \max(0, \min(1, \dfrac{T_{thresh} - T}{T_{range}}))$ <br><br> $Q_M = \max(0, s_{DDF} * s_{RF} * (T - T_{thresh}))$ <br><br> $Q_R = min(S_W * s_{DDF} * s_{RF} * (T_{thresh} - T))$ |
| Interception storage | $dS_I/dt = (P_R + P_M) - E_I - P_E$ | $P_E = \max(0, (S_I - I_{max})/dt)$ <br> $P_R = P - P_S$ <br> $E_I = \min(E_P, S_I/dt)$ |
| Root zone storage | $dS_R/dt = P_E - E_R - Q_R - Q_{perc} + Q_{cap}$ | $Q_{R,\,direct} = \max((S_R + P_E - S_{Rmax}); 0.0)$ <br><br> $Q_{R,\,in,net} = Q_{HR} - Q_{R,\,direct}$ <br><br> $\overline{S_R} = S_R/S_{R,max}$ |



| | | |
|---|---|---|
| | | $E_R = \min\Big((E_P - E_I)$ $\cdot \min(\overline{S_R}/L_P, 1), S_R$ $/dt\Big)$ $Q_R = Q_{R, in, net} \cdot \left(1 - (1 - \overline{S_R})^\beta\right)$ $Q_{perc} = Q_{perc, max} \cdot \overline{S_R}$ $Q_{cap} = Q_{cap, max} \cdot (1 - \overline{S_R})$ |
| Fast-responding storage | $dS_F/dt = Q_{RF} - Q_F$ | $Q_{RF} = Q_R \cdot (1 - d_s)$ $Q_F = K_F \cdot S_F^\alpha$ |
| Slow-responding storage | $dS_S/dt = Q_{RS} + Q_{perc} - Q_S - Q_{cap}$ | $Q_{RS} = Q_R \cdot d_s$ $Q_S = K_S \cdot S_S$ $Q_{TOT} = Q_S + \sum_{class=1}^{n} (Q_F) \cdot F_{hrufrac}$ |