# Peer review of "Catchment Response to Climatic Variability: Implications for Root Zone Storage and Stream Flow Predictions"

_EGUsphere, 2024_

## Author Comment (AC1)

*Comment 1:*

*The study nicely integrates large datasets and modeling to elucidate the minor importance of those differences in Sr,max to hydrologic modeling. The paper is clear, well-written, and the figures are compelling. I have a few minor comments (see below), and I also think the authors could discuss further the implications of the simplification they employ to estimate Sr,max.*

Reply:

We thank the reviewer for his thoughtful, detailed and constructive comments and we highly appreciate his positive overall assessment of our manuscript. We will address all reviewer comments in detail here below.

*Comment 2:*

*Regarding the latter, the authors simplify daily actual evapotranspiration (E_a_daily) to be equal to daily potential evapotranspiration scaled by the decadal ratio of actual to potential ET. They then uses a daily water balance to determine the necessary Sr,max to deliver that daily evapotranspiration. Using a constant ratio to convert potential ET into actual, however, does not necessarily reflect the behavior of catchment vegetation. As a counterpoint, one might expect potential ET to be met fully during periods of low E_p (and plentiful water) and actual ET to approach zero during periods of high-demand/drought. Thus, another way that one could determine the requisite Sr,max – as opposed to equations 4 and 5 – would be to simplify the system such that*

   *E_a = E_p if water is available in storage*

   *E_a = 0 when the water in storage is depleted*

   *Find the value of Sr,max such that the (long-term sum of E_a)/(long-term sum of E_p) equals that desired long-term ratio*

*Such an approach may better represent vegetation response (albeit a little extreme, along the lines of Milly, 1994), and would be more consistent with the complementary hypothesis for evaporation (see multiple references by Szilagyi)*

*It may be that the resulting Sr,max does not differ much from that determined from equations 4 and 5, due to the self-limiting process of ET (e.g., whether one removes 5 mm on day one and then zero on day two or 2.5 mm on day 1 and another 2.5 mm on day 2 may not matter). However, it would be interesting to compare and to see if there is a difference, especially for the monthly/seasonal results, where the differences may have an even larger effect.*

*I understand this may be beyond the scope of the paper. Nevertheless, given the significance of equations 4 and 5 on the central message of this paper, I recommend that the authors spend more time discussing those simplifications, alternative simplifications (such as that above), and the potential implications on the results and conclusions.*

Reply:

This is indeed an import comment and a valid observation. We completely agree that the assumption of a constant ratio $E_A/E_P$ may introduce uncertainties. In particular, during dry periods, this assumption does not account for vegetation water stress and may therefore lead to overestimation of $E_A$ and a potential resulting

inflation of $S_{r,max}$. We will discuss this and the related limitations of the method in more detail in the revised version of the manuscript

*Comment 3:*

*Relatedly, I think the abstract and discussion would benefit from additional acknowledgment that the catchments used in this study are both snow-free and relatively aseasonal. Thus, the conclusions may not be extensible to snow-dominated watersheds and/or those with strong seasonality, such as a Mediterranean climate.*

Reply:

Good point. We agree that many of the study catchments, and in particular those in the Meuse basin, for which we have implemented the hydrological model, are characterized by relatively little snow and little precipitation seasonality. We also agree that the effects of changing $S_{r,max}$ may thus be more pronounced in other environments. We will make this more explicit in the discussion of the revised manuscript.

*Comment 4:*

*Given the nature of the datasets used, I think a more representative title for this work would be "Catchment response to climatic variability: Implications for root zone storage and streamflow predictions." The CAMELS datasets are catchment-based, and the authors are not isolating specific vegetation responses.*

Reply:

Agreed, we will adjust the title to better reflect the analysis.

*Comment 5:*

*I found it somewhat confusing that the meanings of the subscripts modifying evapotranspiration (E) and aridity index (I) were not consistent. When A was used as a subscript, it meant "actual" when modifying evapotranspiration; however, it meant "aridity" when modifying the index, which – in turn – meant it signified potential (not actual) ET. Thus, I_A was not the analog to E_A; rather I_E was the analog to E_A. Perhaps I_A could be used to indicate the evaporative index based on actual ET, whereas I_P could indicate the evaporative index based on potential ET.*

Reply:

We agree with the reviewer that this could potentially be perceived as inconsistency by some. We would nevertheless prefer to keep it as is, as these symbols are frequently found in literature.

*Comment 6:*

*Line 52-60: The authors present their methods of determining Sr,max from a daily water balance (see above). In essence, the Sr,max is the storage volume needed to ensure that daily ET can be met. However, that value represents a minimum value for Sr,max, which – of course – could be larger. It might be worth*

*a comment to that effect, especially since those values of Sr,max are then used in a hydrologic model with a very different mathematics.*

Reply:

This is again a very sharp observation and we of course completely agree. We will explicitly mention this in the revised version of the manuscript.

*Comment 7:*

*Lines 165-172: the numbering scheme used in this paragraph does not exactly match the numbering of the methods sections to which it refers.*

Reply:

Thank you for pointing this out. Will be corrected in the revised manuscript.

*Comment 8:*

*Lines 299-307: I particularly appreciate that the authors sought explanatory variables, such as aridity index, for their results. I expected aridity to be a controlling factor, and it was interesting to learn that it was not.*

Reply:

We agree, we were also quite surprised.

*Comment 9:*

*Line 431: dangling phrase, "the more equilibrated scenario A"*

Reply:

Will be corrected in the revised manuscript

*Comment 10:*

*Equation 5: as written, the equation is circular. What should be used as the argument of the inequalities on the RHS is the integral from t0 to t of (P_daily - E_A_daily) dt rather than S_D,j,i(t)*

Reply:

Indeed. We will correct this in the revised manuscript

*Comment 11:*

*The reference for Dralle, et al. 2021 is missing from the reference list*

Reply:

The reference will be added in the revised manuscript.

*Comment 12:*

*I recognize that figure 7 is intended to explain the methodology and not results. Even so, I recommend that the qualitative character of the distributions for delta_I_E reflect the results of this paper. That is, the distribution for scenario A should be narrower than that for scenario B; and the mean for scenario B could even be shifted away from zero (compare Figure 7 and Figure 11). As is, the figure gives the false impression that the uncertainty across all catchments is greater than across the Meuse watershed alone.*

Reply:

Thank you! That is an excellent idea. We will adjust the figure accordingly in the revised manuscript.

---

## Author Comment (AC2)

*Comment 1:*

*This study calculates the differences in I_E and evaluates how these differences influence the estimates of root zone storage capacities. It further examines how uncertainties in root zone storage capacities affect streamflow predictions in hydrological models. To some extent, the manuscript is well-structured, detailed, and presents valuable ideas. However, several concerns need to be addressed. Additionally, inconsistent formatting and grammar errors diminish the quality of the paper. Overall quality needs to be enhanced.*

Reply:

We thank the reviewer for his/her detailed and constructive comments. We highly appreciate the overall positive assessment of our analysis. We will carefully revise the formatting and grammar errors in the revised manuscript.

*Comment 2:*

*Using absolute values of the I_A deviation in Figure 8 cannot effectively reflect the change in I_A. Using percentage changes would better illustrate how I_A changes to reflect multi-decadal climatic variability. Your use of percentage changes in Figure 12 (c) for root zone storage capacity changes is a good approach.*

Reply:

We completely agree with the reviewer that for many purposes, the analysis of relative changes in $I_A$ is more suitable to meaningfully describe the observed pattern. For our analysis we seek to quantify absolute changes in $S_{r,max}$ over time. To achieve this, we need to quantify the absolute changes in $E_A$ (over $I_E = E_A/P$), which in turn depend on changes in absolute values of $I_A = E_P/P$, as dictated by the Tixeront-Fu equation (Eq.1 in the manuscript). We acknowledge that our description of the procedure has not been sufficiently clear in the original manuscript. We will provide a clearer explanation in the revised manuscript.

*Comment 3:*

*Are the values of the aridity index in Figures 1-3 calculated for the entire period? If so, while the values of I_A deviation in Figure 8 are calculated by decades, it might be better to find a consistent way to present I_A and I_A deviation using the same time period (either the entire period or by decades).*

Reply:

Indeed, the aridity index in Figures 1 – 3 is based on the entire study period, to provide the reader with an overall hydro-climatic context. However, we agree with the reviewer that the actual $I_A$ per decade may be interesting to see. We will include such a Figure in the revised manuscript.

*Comment 4:*

*If percentage changes in I_A are small for most catchments, climatic variability is small. Then is the conclusion that hydrological responses, in terms of changes in I_E, root zone storage capacities, and streamflow, are generally minor under changing climatic conditions reliable?*

Reply:

This is an interesting question. With the available past data records, no fully conclusive answer can be given. In our analysis we only draw the conclusion that effects of the _observed past changes_ of $I_A$ remain rather minor. With hypothetically more pronounced changes in $I_A$, it may plausibly assumed that the effects may be more relevant. However, there is at this point little empirical evidence that such more pronounced changes in $I_A$ have occurred elsewhere over the last 120 years as recently demonstrated by Ibrahim et al. (2024; Figures 4 and S1 therein), nor is there evidence that future changes will significantly exceed those of our analysis at least over the next few decades (Jaramillo et al., 2022; Figures 3 and 4 therein). Both of these previous studies show that globally changes in $I_A$ have in the past and will in the future remain well with in the range of $I_A \sim \pm 0.1$ for the vast majority of catchments.

_Comment 5:_

_Related to 1.c: How many catchments exhibit distinct changing climate conditions? Can percentage changes in I_A and I_E by decades effectively reflect that? If the climate changes are small, their impact on root zone storage capacity changes might be less significant._

Reply:

As shown in Figure 8, less than $\sim 5\%$ of the study catchments exhibit a change of $I_A > \pm 0.1$. A comparable pattern can be found for catchments world-wide (Jaramillo et al., 2022; Ibrahim et al., 2024). Indeed, we agree that if changes in climatic conditions are small, changes in $S_{r,max}$ can also be expected to be low. The actual magnitudes of the change in $S_{r,max}$ are exactly what we aim to quantify in our analysis.

_Comment 6:_

_The legends in Figures 1 and 2 should use periods instead of commas, so they should be 0.1 – 0.2, 0.2 – 0.3, etc., not 0,1 – 0,2, 0,2 – 0,3, etc. Additionally, the title of the legends should be "Aridity Index I_A," with the A as a subscript._

Reply:

Indeed! We agree. This will be corrected in the revised manuscript.

_Comment 7:_

_Figure 11 could be removed; the information is clearly conveyed in the text._

Reply:

We agree. We will remove this figure.

_Comment 8:_

_Units of Figure 13 are incorrect._

Reply:

Thank you for pointing this out. Will be corrected in the revised manuscript.

*Comment 9:*

*Lines 332 to 331, do you mean Figure 13?*

Reply:

Yes. We will correct that in the revised manuscript.

*Comment 10:*

*Line 405: The reference Wang et al., 2016 is missing from the list of references. There may be other missing references as well. A comprehensive reference check is recommended.*

Reply:

We will add the reference to the list and carefully check the rest of the list.

*Comment 11:*

*Line 684: two references listed in one line.*

Reply:

Will be corrected.

References:

Ibrahim, M., Coenders-Gerrits, M., van der Ent, R., & Hrachowitz, M. (2024). Catchments do not strictly follow Budyko curves over multiple decades but deviations are minor and predictable. Hydrology and Earth System Sciences Discussions, 2024, 1-27.

Jaramillo, F., Piemontese, L., Berghuijs, W. R., Wang-Erlandsson, L., Greve, P., & Wang, Z. (2022). Fewer basins will follow their Budyko curves under global warming and fossil-fueled development. Water resources research, 58(8), e2021WR031825.

---

## Author Comment (AC3)

*Comment 1:*

*The article presents an analysis of the sensitivity of root-zone storage capacity to climate variability. The article is generally clear. My main concern on this article is that I did not clearly see the novel insights provided by this study. The authors say a few times that their results corroborates past findings on the limits of the Budyko framework. Actually, I found that they should more clearly emphasize what is new in their results compared to past studies.*

Reply:

We highly appreciate the positive overall assessment by Reviewer #2 and thank him/her for the detailed and insightful comments! We will make it clearer what the novelty of this work is. Briefly, the analysis is a direct follow up to the paper by Bouaziz et al. (2022). In their paper, the potential effects of an *assumed, hypothetical future change* in $S_{r,max}$ for stream flow *under a projected future climate* where explored. The novelty here is that we do not assume but instead actually *quantify for the first time past changes* of $S_{r,max}$ based on historical observations and use them to explore the effects thereof on modelled stream flow in the past. In comparison to Bouaziz et al. (2022), whose analysis remained hypothetical, these two new points allow to base the analysis of $S_{r,max}$ changes on *real world observations* and to *evaluate the modelled effects* of changing $S_{r,max}$ on modelled stream flow *against observed stream flow*.

We will clarify that in the revised manuscript.

*Comment 2:*

*L67-69: What about groundwater exchanges? Should not they be considered also?*

Reply:

We agree. Ideally, groundwater exchange fluxes should be considered. However and unfortunately, it remains problematic if not at all impossible to meaningfully quantify these fluxes with current observation technology. This is, however, a limitation of the *vast majority* of hydrological studies and not limited to our analysis (e.g. Condon et al., 2020). To keep the uncertainties introduced by potential groundwater exchange fluxes as low as possible, we have therefore excluded all catchments that show clear evidence of a water balance deficit or surplus, which both indicate groundwater exchange (e.g. Bouaziz et al., 2018), from our analysis. This was done by discarding all catchments that plotted (1) above the upper limit of $E_A/P$ ($E_A/P > E_P/P$ and $E_A/P > 1$) and thus outside the physically plausible realm in the Budyko framework (e.g. Fig.3), thereby indicating groundwater export and (2) more than 0.25 below the analytical Budyko solution, indicating potential groundwater import.

*Comment 3:*

*Please explain in simple words what low or high values of omega mean in terms of water balance type. Say omega should be positive.*

Reply:

The parameter ω in the parametric Tixeront-Fu equation (Eq.1 in the original manuscript) can range between 1 and ∞ (e.g. Zhang et al., 2001; Greve et al., 2015; Andreassian et al., 2016). The lower the ω of a catchment the lower $E_A/P$ of this catchment (and inversely, the higher Q/P). We will add this in the revised manuscript.

*Comment 4:*

*L141: At this stage of the article, it is unclear why this specific PE formula was used. Maybe this could be justified in a few words (even if some comments are later provided in the discussion section).*

Reply:

Agreed. We will add an explanation in the Methods section.

*Comment 5:*

*L151-154: I do not understand why apparently "leaking" catchments where excluded from the analysis. There are probably many catchments in the remaining dataset, which exhibit groundwater exports even if there are within the limits of the graph. The limit IE>IA is arbitrary and does not really corresponds to underlying processes (groundwater exports or not). Would the analysis be very different if all the catchment had been kept to conduct the analysis? Does this catchment selection partly explain the apparently very consistent behaviour of UK and US catchments?*

Reply:

*Apparently* leaking catchments were excluded as the estimates of $E_A$ in these catchments cannot be distinguished from groundwater export with the available data in the water balance, i.e. $P - Q = E_A - Q_{GW,export}$. This may, as correctly remarked by the reviewer, of course also affect catchments within the limits. However, as with the available data there is no meaningful way to quantify groundwater export (or import) otherwise, the best that we can do is to at least exclude those catchments that plot outside the *physically possible* realm and for which we actually *know* that groundwater export plays a role. As the number of catchments that plot outside the $I_E > I_A$ limit remains very low($< 1\%$), the effect on the overall results of analysis are very minor. As such it can also not explain the consistent behaviour of the UK and US catchments. We will discuss this in some more detail in the revised version of the manuscript.

*Comment 6:*

*Section 4: I felt a bit confused in reading the results section. In the first part of the analysis, the Meuse basin seems to be somehow outlier in its behaviour compared to the UK and US datasets, but no clear explanations on this behaviour is found. Therefore the added value of this small catchment sub-set in this part of the analysis is unclear. In the second part where the process-based model is applied, only the Meuse basin is used. Though I understand it is difficult to apply such a model on large datasets, I found it makes the study less clear and the overall conclusions more difficult to draw.*

Reply:

We appreciate that the reviewer points out that our choices are not completely clear. Briefly, we have included the catchments of the Meuse basin for two reasons: (1) as described in our reply to *Comment 1* above, this analysis is a direct follow up to the work of Bouaziz et al. (2022), who investigated the effects of future $E_A$ (using $I_E = E_A/P$) on $S_{r,max}$ and the associated stream flow. Their study was based on *projections of future climate* and the *assumption* that catchments largely remain on their parametric Budyko curve defined by a constant parameter ω. They explored future effects and could therefore in their study not test

whether these assumptions hold. As the Bouaziz et al. (2022) analysis was executed in the catchments of the Meuse basin, we here decided to zoom in on these same catchments to allow a direct comparison with that previous study. (2) Future estimates of $E_A$, and thus by extension also future estimates of $S_{r,max}$, depend on the deviation of catchments from their specific parametric Budyko curve, i.e. $\Delta I_E$ over time. To our knowledge, there has so far been no systematic analysis to quantify these distributions of deviations $\Delta I_E$ over time. In other words, we do not know to which extent the assumption of Bouaziz et al. (2022) that catchments move along a specific curve over time actually holds. We have therefore decided to not only quantify these distributions for the Meuse basin but to provide a wider context of which magnitudes of deviations need to be expected over a larger sample of catchments. The fact that the Meuse catchments showed a rather pronounced change in $I_E$, as visible by the skewed distribution of $\Delta I_E$, while the samples of catchments elsewhere were characterized by much lower changes, on average, gave us the opportunity to quantify the effects of both, extreme and average changes, on $S_{r,max}$ and thus on predicted stream flow in the Meuse basin. Please note, that here we did not seek to identify the reasons for the pronounced $\Delta I_E$ in the Meuse basin. This was previously explored by Fenicia et al. (2009), who found that changes in land use management are the most likely reason for the observed pattern. Instead, our objective was to explicitly quantify the change and to analyse which knock-on effects it has on $S_{r,max}$ and eventually on stream flow predictions. We will make this clearer in the revised version of the manuscript.

*Comment 7:*

*Section 4.1: I found figures 9 to 11 not very useful. Maybe the authors could say what were their results without showing figures (or putting them in SM).*

Reply:

We agree. We will condense some of these figures and show them in the Supplementary Material.

*Comment 8:*

*Sections 4.3.2 and 4.3.3: I found these parts of the article tedious to read. A lot of detailed information is given, which makes it difficult to draw the overall picture.*

Reply:

Thank you for this observation. We will rework these sections and make them more concise and readable.

*Comment 9:*

*Conclusion: see main comment above*

Reply:

Agreed, we will make it clearer in the Conclusion what the novel findings of our analysis are

*Comment 10:*

*14: I found this figure not very useful (overall I found there are too many figures in this article and their number could probably be reduced).*

Reply:

We agree. We will move this figure to the Supplementary Material.

*Comment 11:*

*The reference Hulsman et al. (2021) is missing in the list of references.*

Reply:

Thank you for pointing this out. We will add the reference.

*Comment 12:*

*UK is used in notations, but GB is used for the CAMELS dataset. Maybe use a single abbreviation to be consistent.*

Reply:

We absolutely agree. This will be corrected.

References:

Bouaziz, L. J., Aalbers, E. E., Weerts, A. H., Hegnauer, M., Buiteveld, H., Lammersen, R., ... & Hrachowitz, M. (2022). Ecosystem adaptation to climate change: the sensitivity of hydrological predictions to time-dynamic model parameters. Hydrology and Earth System Sciences, 26(5), 1295-1318.

Fenicia, F., Savenije, H. H. G., & Avdeeva, Y. (2009). Anomaly in the rainfall-runoff behaviour of the Meuse catchment. Climate, land-use, or land-use management?. Hydrology and Earth System Sciences, 13(9), 1727-1737.

---

## Author Response (AR2)

REVIEWER #1

Author note:

Adjustments in response to the comments of Reviewer #1 are highlighted in yellow in the revised manuscript

*Comment 1:*

*The study nicely integrates large datasets and modeling to elucidate the minor importance of those differences in Sr,max to hydrologic modeling. The paper is clear, well-written, and the figures are compelling.*
*In this revision, the authors have addressed the comments of the reviewers, and the paper is improved. The writing could use another pass to ensure appropriate grammar throughout, particularly subject-verb agreement and proper use of commas. I recommend acceptance, and I offer a couple of additional suggestions that may enhance the paper.*

Reply:

We thank the reviewer for his positive assessment of our manuscript. In the revised version we have done a detailed grammar check and reworded incorrect expressions.

*Comment 2:*

*Lines 362-366 - The authors note the large range of delta_Sr,max,exp for catchments with aridity indices between 0.75-1.0 and offer that it may be due to the large number of catchments within that aridity range. In looking at figure 10, it also appears that the catchments with large, positive delta_Sr,max,exp are also those with deep roots (300-400+ mm); thus, the relative difference is smaller and consistent with catchments that are drier and wetter. This may be worth a note.*

Reply:

This is indeed a sharp observation. We have added a sentence to highlight this effect.

*Comment 3:*

*A central claim of the work is that the variations in delta_I_E and delta_Sr,max are small. To provide additional context for that claim, it might be helpful to report the variability in I_E and Sr,max across the catchments. That would help the reader understand how the variability through time (delta_x) compares to the variability across space (i.e., the variability from catchment to catchment is much greater than for a single catchment through time). This would speak to the value/utility of using a constant omega (based on historic data) in the Fu-Budyko representation for prediction.*

Reply:

This is a good idea. We have added an additional figure plotting $\Delta I_A$ against $\Delta I_E$ in the Supplementary Material to demonstrate that effect.

Author note:

Adjustments in response to the comments of Reviewer #2 are highlighted in green in the revised manuscript

*Comment 1:*

*I thank the authors for their detailed reply to the review comments and for the modifications made in the article. I think they clarify the points that were found unclear by the two reviewers.*
*I have a small disagreement with the authors on their reply on catchment selection based on water balance considerations. I think that there are many catchments which have an unusual water balance for natural reasons and which should not be discarded from the samples used to test modelling approaches. Keeping these catchments sometimes yields lower average performance but also gives information on the suitability of the tested methods on more varied types of catchments. However I think this would not significantly change the overall conclusions of this study, so I do not require further modifications on this aspect.*

Reply:

We thank the reviewer for the positive assessment of our manuscript.

*Comment 2:*

*Please just homogenize the way streamflow is written (stream flow or streamflow).*

Reply:

We have homogenized the spelling and now use "stream flow" throughout the manuscript.